# Have precipitation extremes and annual totals been increasing in the world's dry regions over the last 60 years?

Sebastian Sippel[1,2], Jakob Zscheischler[2], Martin Heimann[1], Holger Lange[3], Miguel D. Mahecha[1,4,5], Geert Jan van Oldenborgh[6], Friederike E. L. Otto[7], and Markus Reichstein[1,4,5]

[1]Max Planck Institute for Biogeochemistry, Jena, Germany
[2]Institute for Atmospheric and Climate Science, ETH Zürich, Zürich, Switzerland
[3]Norwegian Institute of Bioeconomy Research, Ås, Norway
[4]German Centre for Integrative Biodiversity Research (iDiv), Leipzig, Germany
[5]Michael Stifel Center Jena for Data-Driven and Simulation Science, Jena, Germany
[6]Weather and Climate Modeling, Koninklijk Nederlands Meteorologisch Instituut, De Bilt, Netherlands
[7]Environmental Change Institute, University of Oxford, South Parks Road, Oxford, United Kingdom

*Correspondence to:* Sebastian Sippel (ssippel@bgc-jena.mpg.de)

**Abstract.** Daily precipitation extremes and annual totals have increased in large parts of the global land area over the last decades. These observations are consistent with theoretical considerations of a warming climate. However, until recently these trends have not been shown to consistently affect dry regions over land. A recent study, published by Donat et al. (2016), now identified significant increases in annual-maximum daily extreme precipitation (Rx1d) and annual precipitation totals (PRCP-
TOT) in dry regions. Here, we revisit the applied methods and explore the sensitivity of changes in precipitation extremes and annual totals to alternative choices of defining a dry region (i.e., in terms of aridity as opposed to precipitation characteristics alone). We find that a) statistical artifacts introduced by data preprocessing based on a time-invariant reference period lead to an overestimation of the reported trends by up to 40%, and that b) the reported trend of globally aggregated extremes and annual totals are highly sensitive to the definition of a 'dry region of the globe'. For example, using the same observational
dataset, accounting for the statistical artifacts and based on different aridity-based dryness definitions, we find a reduction in the positive trend of Rx1d from the originally reported +1.6% decade$^{-1}$ to +0.2 to +0.9% decade$^{-1}$ (period changes for 1981-2010 averages relative to 1951-1980 are reduced to -1.32 to +0.97% as opposed to +4.85% in the original study). If we include additional but less homogenized data to cover larger regions, the global trend increases slightly (Rx1d: +0.4 to +1.1% decade$^{-1}$), and in this case we can indeed confirm (partly) significant increases in Rx1d. However, these globally aggregated
estimates remain uncertain as considerable gaps in long-term observations in the Earth's arid and semi-arid regions remain. In summary, adequate data preprocessing and accounting for uncertainties regarding the definition of dryness are crucial to the quantification of spatially aggregated trends in precipitation extremes in the world's dry regions. In view of the high relevance of the question to many potentially affected stakeholders, we call for a well-reflected choice of specific data processing methods and the inclusion of alternative dryness definitions to guarantee that communicated results related to climate change be
robust.

# 1 Introduction

Daily precipitation extremes are expected to increase over large parts of the global land area roughly by 6-7% per °C of warming due to a higher atmospheric water-holding capacity as specified by the Clausius-Clapeyron equation (Allen and Ingram, 2002; Trenberth et al., 2003). Quantifying and predicting changes in precipitation characteristics due to climate change is crucial for water availability assessments and adaptation to climate change (IPCC, 2012; Greve et al., 2014). On a global scale, daily precipitation extremes have been observed to intensify (Donat et al., 2013a; Westra et al., 2013; O'Gorman, 2015), consistent with global model simulations (Fischer and Knutti, 2015), and coincide with a global-scale increase in observed annual precipitation totals (Donat et al., 2013a). However, there is little information to date on how precipitation characteristics have changed in the past over dry land areas and how they will change in the future. Donat et al. (2016) investigate whether and to what extent daily precipitation extremes (Rx1d) and annual precipitation totals (PRCPTOT) have increased over the last 60 years using observational data. The authors identify rapid increases in Rx1d over dry regions, which strongly outpace the corresponding increases over wet areas, and find a similar pattern for PRCPTOT.

The question whether precipitation extremes increase in dry regions is highly relevant in the context of climate change adaptation, as generally dry areas may be less prepared to deal with precipitation extremes (Ingram, 2016). Consequently, the recent report on increasing Rx1d in dry areas was highlighted in major Science journals (including *Nature* (Tollefson, 2016) and *Nature Climate Change* (Ingram, 2016)) and received a lot of media coverage [1, 2, 3, 4, 5, 6], which indicates the importance of this topic for the scientific community, the public and decision makers.

However, scrutinizing the findings by Donat et al. (2016) reveals two major issues of concern: Firstly, the applied statistical approach introduces two systematic biases that lead to a substantial overestimation of the increase in PRCPTOT and Rx1d of up to 40% in dry regions. Wet regions, by contrast, are only affected to a limited degree due to an approximate cancellation of errors in trend estimates. Secondly, the definition of a dry region used in Donat et al. (2016) based on PRCPTOT and Rx1d alone does only partly reflect the water balance and thus water availability (for instance, it ignores losses through evapotranspiration). Furthermore, defining dryness based on low Rx1d (Donat et al., 2016) fells a decision on whether a region is dry or not based on only one day in the year. The chosen definitions thus induce considerable uncertainty in the reported results. If we test alternative but well-established definitions of a 'dry region' (based on water supply and demand, either implicitly or explicitly, cf. Köppen, 1900; Greve et al., 2014) and apply the appropriate statistical tools, we find strongly increasing trends and period changes (1981-2010 averages relative to the 1951-1980 reference period) in PRCPTOT and Rx1d in the world's dry regions. An accurate quantification of trends and changes in precipitation characteristics is of high relevance and a crucial prerequisite in the context of making climate change adaptation decisions (e.g. IPCC, 2014).

---

[1]http://www.huffingtonpost.com/entry/global-warming-will-bring-extreme-rain-and-flooding-study-finds_us_56e081c7e4b0860f99d796ab

[2]https://www.theguardian.com/environment/2016/mar/08/hotter-planet-spells-harder-rains-to-come-study

[3]https://www.sciencedaily.com/releases/2016/03/160308105625.htm

[4]http://phys.org/news/2016-03-global-world-driest-areas.html

[5]http://www.abc.net.au/news/2016-03-08/climate-change-could-bring-more-rain-to-deserts-study/7229236

[6]http://www.asce.org/magazine/20160412-climate-change-to-cause-more-precipitation-in-dry-regions,-researchers-say/

## 2 On data pre-processing based on a time-invariant reference period

As a first step in the analysis of Donat et al. (2016), the authors normalise the 60-year time series in the gridded HadEX2 dataset (Donat et al., 2013a) for each grid point with the sample mean of a 30-year reference period (1951-1980), which is a widespread procedure in climate science. However, this procedure artificially increases the mean of the spatial distribution in the out-of-base period (1981-2010) in all investigated time series, simply because variability in the sample means inflates the signal in the latter period (Sippel et al., 2015). To illustrate this point, consider two hypothetical climate regions of the same size: In region one, the mean of a precipitation quantity increases between two periods (from 100 to 200mm, say), for example due to a few large extremes, whereas it decreases by exactly the same amount in region two (i.e. from 200 to 100mm). Consequently, over the combined period the spatial average and the spread of the two regions would be statistically indistinguishable. However, normalising by the mean of the first time period would imply that the spatial average across both regions for the second period is 1.25 (the average of 0.5 and 2), i.e. a spurious increase of 25% between both periods. This issue is illustrated in Fig. 1 for an artificial dataset that consists of $n = 10^4$ time series (e.g., 'grid cells') that are drawn randomly and independently from a Generalized Extreme Value distribution (GEV, Coles et al., 2001). The GEV distribution provides an asymptotical limit model for maxima derived from a sequence of random variables with fixed block size (Coles et al., 2001, e.g. Rx1d,), and is therefore appropriate to illustrate this issue. Normalising each time series in the artificial dataset by its mean in the first period yields a spatial 'reference period distribution' that is different from the spatial 'out-of-base period distribution' (and from the original GEV distribution, Fig. 1a). In particular, this normalisation leads to increased spatial averages in the out-of-base period (Fig. 1b). Furthermore, the normalisation procedure induces a considerable increase in the variance, skewness and higher statistical moments in the spatial distribution in the out-of-base period (see e.g. Fig. 1a), which would be of relevance if higher statistical moments (changes in spatial variance, etc.) were studied. The reason for this difference lies in the fact that the estimated sample means (of the reference period) are statistically dependent to reference period time series, but (virtually) independent to the time period that lies outside of the reference period (Zhang et al., 2005; Sippel et al., 2015). It is worth noting that these biases can be understood analytically (Appendix A). The expected value $\Delta_{\text{bias}}$, defined as the relative bias in the out-of-base period, can be well approximated for each grid cell with

$$\Delta_{\text{bias}} \approx \frac{\sigma^2}{\mu^2 n_{\text{ref}}}, \tag{1}$$

where $\mu$, $\sigma$, and $n_{\text{ref}}$ denote the time series' mean, standard deviation, and reference period length, respectively (Appendix A). Thereby, it can immediately be seen that the introduced bias is systematically positive outside of the reference period, and it is proportional to the ratio of $\frac{\sigma^2}{\mu^2}$ for any fixed reference period length.

An additional statistical bias stems from the choice of the world's 30% wettest and 30% driest regions based on the climatology of PRCPTOT and Rx1d in the reference period (1951-1980). Because 30 years are fairly short to derive a robust climatology of the tails of the precipitation distribution, the computed changes in wet and dry regions are distorted by the 'regression to the mean' phenomenon (Galton, 1886; Barnett et al., 2005). To illustrate this issue, recall the conceptual two-region example quoted above, where variation between the two available time periods would be entirely due to random causes. If any of the two periods would be chosen to stratify the dataset in one dry and one wet region, this would result in opposing changes

(i.e. dry gets wetter, wet gets drier) in the independent period. In other words, selecting from the dry (wet) end of the spatial distribution in one subset of the dataset, or 'reference period', will result in a higher probability for wetter (drier) conditions in the remaining years if any type of random variation plays a role (Table 1, and Fig. 2 for changes due to both statistical effects). Although random variations in 30-year averages are not very large (compare Fig. 3a,b and Fig. 3c,d), it is important to consider

this effect as it is indeed noticeable in the reported results (Table 1).

The chosen normalisation approach combined with the spatial point selection method results in a bias toward PRCPTOT and Rx1d increasing at a faster rate in dry regions compared to wet regions. Over dry regions, both effects lead to an overestimation of the trends in precipitation totals and extremes by +40.3% and +33.2% (+32.9% and +40.4% overestimation in the reported period changes from 1951-1980 to 1981-2010), respectively (Fig. 2, Table 1). In contrast, in wet regions both errors roughly

cancel each other out in the case of extremes (increase by only +8.7%) and lead to a small underestimation of the increase in total precipitation (-13.7%). In summary, we find that the applied preprocessing steps are crucial to accurately quantify changes in precipitation extremes and annual totals. In the study under scrutiny, if the dryness definition is kept, trends and period increments are corrected to much lower values, but the trends and period increments remain positive and significant (see Fig. 2).

## 3   On the definition of a dry region

Climatological dryness is typically not determined by water supply alone but also depends on atmospheric water demand, that is the ability to evaporate water from the land surface (Köppen, 1900). This means that "*we cannot tell whether a climate is moist or dry by knowing precipitation alone; we must know whether precipitation is greater or less than potential evapotranspiration*", as Charles Warren Thornthwaite put it in a landmark paper (Thornthwaite, 1948); a statement that is indeed

mirrored in present-day literature (e.g. Hulme, 1996; Cook et al., 2004; Feng and Fu, 2013; Greve et al., 2014; Sherwood and Fu, 2014; Huang et al., 2015), and international reports (Middleton and Thomas, 1992; Millennium Ecosystem Assessment, 2005; Adeel et al., 2005). Metrics and indicators that are typically used to determine climatological dryness and changes therein are derived from this concept, e.g. the aridity index as the ratio of precipitation to potential evapotranspiration (e.g. Hulme, 1996; Greve et al., 2014; Milly and Dunne, 2016). However, in other studies dry regions are defined based on monthly or

annual precipitation totals (Allan et al., 2010; Sun et al., 2012; Liu and Allan, 2013). Donat et al. (2016) define dry regions for the PRCPTOT analysis based on low annual precipitation totals, and dry regions for the Rx1d analysis are based on moderate annual-maximum daily precipitation. Consequently, this latter definition fells a decision whether a region is dry or not based on the precipitation amount of a single day per year. Regions in northern Europe such as parts of Scandinavia or the Netherlands fall in the 'dry' class because of relatively small annual-maximum daily precipitation (Fig. 3). Hence, different notions of what

constitutes a dry region can contrast each other, resulting in regions being dry in one definition and wet in another (e.g., parts of Northeastern Europe, Fig. 3). These variation in dryness definitions consequently induces uncertainties in the interpretation of changes in precipitation extremes and totals in the 'world's dry regions'. These definition-related differences can be substantial - for example, as much as 50.8% (PRCPTOT) and 71.8% (Rx1d) of the 'dry grid cells', following the respective definitions in

Donat et al. (2016), are neither arid nor semi-arid (Appendix B, Figure B1), and would thus not be considered dry if a definition based on both water supply and atmospheric demand were to be used.

To clarify this issue, we test the sensitivity of the reported increases in Rx1d and PRCPTOT to the choice of dryness definition by using a variety of different dryness definitions (Fig. 3). Hence, we evaluate trends and period increments in Rx1d and PRCPTOT in

1. regions that fall below the global 30% quantile in HadEX2 in the respective diagnostic (Rx1d or PRCPTOT), following Donat et al. (2016),

2. dry regions ('B-climates') from a traditional climate classification based on temperature and precipitation (Köppen, 1900; Kottek et al., 2006),

3. dry regions as identified from an aridity-based definition of dryness (Greve et al., 2014), and

4. dry and transitional regions combined from the latter definition (Greve et al., 2014).

In addition, we test uncertainties related to the temporal coverage of the dataset by relying on time series with at least 90% coverage (cf. Donat et al., 2016) and furthermore also analyse only time series without missing values (100% coverage).

Our results show that, if dry regions are defined based on water availability (i.e., dry regions following either Greve et al. (2014) or Köppen (1900)) and statistical artefacts are accounted for, in dry or dry and transitional regions combined, the trends reduce from the originally reported 1.6% decade$^{-1}$ (2.0% decade$^{-1}$) to +0.2 to +0.9% decade$^{-1}$ (+0.0 to +1.2% decade$^{-1}$) for Rx1d (PRCPTOT), respectively. The uncertainty range reflects the choice of aridity mask used and the temporal coverage of the time series considered (see Table 2 and Table 3). Similarly, period changes between 1951-1980 and 1981-2010 would be reduced to -1.32 to +0.97% (+0.5 to +3.8%) as opposed to +4.85% (+6.3%) for Rx1d (PRCPTOT) in the original study. Although the trends remain positive, based on a two-sided Mann-Kendall test, no significant trends in Rx1d and PRCPTOT can be detected in the world's dry regions. However, the coverage of the world's arid regions with long-term observational monitoring data is rather sparse and largely confined to arid and semi-arid regions in North-America and Eurasia (Fig. 3), and thus large uncertainties remain. A few of the data gaps in HadEX2 in arid and semi-arid regions can be filled with available data from the less homogenized GHCNDEX dataset (Donat et al., 2013b, Appendix B, Figure B2). In the dry (Köppen, 1900; Greve et al., 2014) and dry-transitional regions (Greve et al., 2014) of this merged dataset, the magnitude of the trends and period changes remains largely the same for Rx1d (trends: +0.4 to +1.1% decade$^{-1}$, period changes: -0.16 to +1.41%), but with now more significant p-values due to a higher data coverage (Table 2). For PRCPTOT, the HadEX2-GHCNDEX merged dataset reveals on average increased and significant trends (+0.6% to +1.9% decade$^{-1}$) and period changes (+1.7% to +5.1%). The reported results are consistent with earlier studies that report modest increases in Rx1d and PRCPTOT in predominantly arid and semi-arid subsidence regions based on model simulations (Kharin et al., 2007; Fischer and Knutti, 2015) and in observations for individual subtropical regions such as Australia or the Mediterranean (Westra et al., 2013; Lehmann et al., 2015). If 'the world's dry regions' are defined based on falling below a global 30% threshold in Rx1d or PRCPTOT in the HadEX2 dataset (Donat et al., 2016), we indeed confirm robust increases in both Rx1d and PRCPTOT. Thus, the originally

reported robust increases in both diagnostics are highly sensitive to the definition of a 'dry region', and appear to stem from regions with relatively moderate extreme (Rx1d) or average (PRCPTOT) precipitation, such as regions in Northern Europe (Rx1d, Fig. 3) or North-East Siberia (PRCPTOT, Fig. 3).

## 4 Conclusions

Monitoring and an accurate quantification of trends in meteorological risks in a rapidly changing Earth system is a prerequisite to well-informed decision-making in the context of climate change adaptation (IPCC, 2014). In this context, short reference periods that are defined on a subset of the available dataset for normalisation or data preprocessing purposes should be avoided, as this procedure inevitably introduces biases (Zhang et al., 2005; Sippel et al., 2015). In the present study under scrutiny, these statistical effects reduce the reported trends and period changes by up to 40%, but the direction of the overall signal remains unchanged (i.e. increasing trends in Rx1d and PRCPTOT in regions of moderate extreme precipitation and low annual totals, respectively).

Furthermore, the definition of a 'dry region' induces considerable uncertainty in quantifying changes in Rx1d and PRCPTOT in such areas. If dryness is defined based on water supply and demand (i.e. aridity), we find much smaller trends and period increments in Rx1d and PRCPTOT, which are almost exclusively positive but in many cases insignificant (Table 2 and Table 3). Hence, overall we can confirm an indication towards increases in both metrics in the world's dry regions. However, it is important to stress that many of the world's dry regions, such as large arid and semi-arid regions in Africa, the Arabian Peninsula, and partly South America are not covered by monitoring datasets that are available at present. This fact highlights the importance of consistent, long-term monitoring efforts, data quality control, development and maintenance of long-term datasets (Alexander et al., 2006; Donat et al., 2013a, b), and also emphasises that the results reported here should be regarded as indicative only for those arid regions where data is available.

In summary, understanding and disentangling on-going changes in precipitation characteristics in the world's dry regions remains a research priority of high relevance. In this context, our paper demonstrates that 1) data preprocessing can introduce substantial bias, and 2) trends and period changes can be sensitive to the specific choice of dryness definition that is used; therefore we urge authors to be considerate and specific regarding both choices and to consider associated uncertainties.

## Appendix A: Analytical approximation of the expected value for the normalisation-induced bias

Assumptions and Notation:

- Assume independent and identically distributed (i.e., stationary) variables $X_{t,i}$ with mean given by $\mathbf{E}(X) = \mu$ and variance $\mathbf{Var}(X) = \sigma^2$. Let the subscripts $t$ and $i$ denote time and grid cell index, respectively. Note that in real-world applications, the biases could be estimated analytically by allowing for different sample means and variances across space.

- Let $t_{oob}$ be an arbitrary time step in the 'out-of-base' (independent) period, and $t_{ref}$ as an arbitrary time step inside the reference period. Let $n_{ref}$ denote the length of the reference period.

- Let $\Delta_{bias} = \mathbf{E}(\frac{X_{t_{oob},i}}{\hat{\mu}_{ref,i}}) - 1$ denote the relative change induced by normalisation by the mean of an independent reference period (i.e., 'normalisation bias', $X_{t_{oob},i}$ is not contained in $\mu_{ref,i}$).

Our objective is to find an analytical approximation of the expected value for the artificially induced relative change ($\Delta_{bias}$) by dividing a random variable $X_{t_{oob},i}$ as defined above by a sample mean estimated from different samples ('reference samples?') drawn from the same distribution ($\hat{\mu}_{\text{ref},i} = \frac{1}{n} \sum_{t_{\text{ref}}=1}^{n_{\text{ref}}} X_{t_{\text{ref}},i}$, where $\mathbf{E}(\hat{\mu}_{\text{ref},i}) = \mu$), i.e.

$$\Delta_{bias} = \mathbf{E}(\frac{X_{t_{oob},i}}{\hat{\mu}_{\text{ref},i}}) - 1 \approx f(\mu, \sigma, n_{\text{ref}}). \tag{A1}$$

Clearly, for large $n_{\text{ref}}$ this quantity should go to 0. Because $X_{t,i}$ and $\hat{\mu}_{\text{ref},i}$ are independent, we can write,

$$\Delta_{bias} = \mathbf{E}(X_{t,i})\mathbf{E}(\frac{1}{\hat{\mu}_{\text{ref},i}}) - 1 = \mu\mathbf{E}(\frac{1}{\hat{\mu}_{\text{ref},i}}) - 1. \tag{A2}$$

If we subsitute $\hat{\mu}_{\text{ref},i} = \mu(1 + \epsilon_{\text{ref},i})$, where $\mathbf{E}(\epsilon_i) = 0$, $\mathbf{Var}(\epsilon_i) = \frac{\sigma^2}{\mu^2 n_{\text{ref}}}$ (because $\epsilon_{ref,i} = \frac{\hat{\mu}_{ref,i}}{\mu} - 1$, and $\mathbf{E}(\hat{\mu}_{ref,i}) = \mu$ and $\mathbf{Var}(\hat{\mu}_{ref,i}) = \frac{\sigma^2}{n_{\text{ref}}}$), and the subscript ref has been dropped from $\epsilon_i$ for convenience, we get

$$\Delta_{bias} = \mu\mathbf{E}(\frac{1}{\mu(1 + \epsilon_i)}) - 1 = \mathbf{E}(\frac{1}{1 + \epsilon_i}) - 1. \tag{A3}$$

A Taylor expansion around the function $g(x) = \frac{1}{1+x}$ at $x = 0$ yields

$$g(x) = \frac{1}{1 + x} = 1 - x + x^2 - x^3 + x^4 - x^5 + ... \tag{A4}$$

We will see below that the convergence criterion $\epsilon_i < |1|$ of the Taylor series is met in practically relevant cases, but it should be noted that convergence cannot be ensured in all theoretically conceivable cases. Using Taylor expansion, $\Delta_{bias}$ can be approximated, making use of the linearity of the expectation operator $\mathbf{E}()$ and of the fact that $\mathbf{E}(\epsilon_i) = 0$ and $\mathbf{E}(\epsilon_i^2) = \mathbf{Var}(\epsilon_i) = \frac{\sigma^2}{\mu^2 n_{\text{ref}}}$ by definition,

$$\Delta_{bias} = \mathbf{E}(\frac{1}{1 + \epsilon_i}) - 1 \tag{A5}$$

$$= \mathbf{E}(1 - \epsilon_i + \epsilon_i^2 - \epsilon_i^3 + \epsilon_i^4 - \epsilon_i^5 + ...) - 1 \tag{A6}$$

$$= \frac{\sigma^2}{\mu^2 n_{\text{ref}}} - \mathbf{E}(\epsilon_i^3) + \mathbf{E}(\epsilon_i^4) - \mathbf{E}(\epsilon_i^5) + ... \tag{A7}$$

This expression yields a sum over the central moments of the distribution of $\epsilon_i$'s. For a symmetric probability distribution (recall that $\epsilon_i$ denote the deviations of the sample means in a reference period around the underlying true mean) $E(\epsilon_i^k) = 0$, where $k$ is any uneven exponent $k \in \mathbb{N}$. Eq. A7 then reduces to

$$\Delta_{bias} = \frac{\sigma^2}{\mu^2 n_{\text{ref}}} + \mathbf{E}(\epsilon_i^4) + \mathbf{E}(\epsilon_i^6) + ... \tag{A8}$$

As long as $\epsilon_i < |1|$ is fulfilled, the quadratic term dominates both Eq. A7 and Eq. A8. The present analytical approximation (both Eq. A7 and Eq. A8) provides the important insights that 1) normalisation with a 'reference period sample mean' leads to an artificial increase of spatial averages in the out-of-base period, i.e. the bias is always positive in the out-of-base period, $\Delta_{bias} > 0$, and 2) that $\Delta_{bias} \propto (\frac{\sigma}{\mu} \frac{1}{\sqrt{n_{\text{ref}}}})^2$, i.e. the square of the coefficient of variation in the distribution of sample means (i.e., $c_v[\hat{\mu}_{\text{ref},i}] = \frac{\sigma}{\mu\sqrt{n_{\text{ref}}}}$). For any fixed $n_{\text{ref}}$, the amplitude of the normalisation-induced biases only depends on the square of the ratio $\frac{\sigma}{\mu}$. We verify below numerically that this approximation works well for random variables $X_{t,i}$ drawn from i. a Gaussian distribution, ii. a Generalized Extreme Value distribution with two different choices for the shape parameter ($\xi = 0$, 'Gumbel distribution', and $\xi \neq 0$).

**Gaussian distribution**

Assume $X_{t,i} \sim \mathcal{N}(\mu, \sigma^2)$, the distribution of the sample mean deviations from the true mean will follow $\epsilon_i \sim \mathcal{N}(0, \frac{\sigma^2}{\mu^2 n_{\text{ref}}})$. If we substitute with $\epsilon_i = \frac{\sigma}{\mu} \frac{1}{\sqrt{n_{\text{ref}}}} Y$, where $Y \sim \mathcal{N}(0,1)$ in Eq. A8, the above expression reduces to

$$\Delta_{bias} = \frac{\sigma^2}{\mu^2 n_{\text{ref}}} + (\frac{\sigma}{\mu} \frac{1}{\sqrt{n_{\text{ref}}}})^4 \mathbf{E}(Y^4) + (\frac{\sigma}{\mu} \frac{1}{\sqrt{n_{\text{ref}}}})^6 \mathbf{E}(Y^6) + ... \tag{A9}$$

Because higher-order moments of a standard normal distributed random variable are well-known and can be derived analytically (Johnson et al., 1994, i.e., $\mathbf{E}(Y^4) = 3$, $\mathbf{E}(Y^6) = 15$), an analytical expression of the normalisation-induced bias becomes straightforward:

$$\Delta_{bias} \approx \frac{\sigma^2}{\mu^2 n_{\text{ref}}} + 3(\frac{\sigma}{\mu} \frac{1}{\sqrt{n_{\text{ref}}}})^4 + 15(\frac{\sigma}{\mu} \frac{1}{\sqrt{n_{\text{ref}}}})^6. \tag{A10}$$

A comparison of Eq. A10 (i.e. the first three terms in the Taylor approximation) to numerical simulations shows that the analytical approximation works well (Fig. A-1a). Furthermore, the estimation of mean and standard deviation from the empirical time series to calculate the expected value for the biases is unbiased and show surprisingly little variation (Fig. A-1b) even for a relatively small number of grid cells, where random variation in stationary time series becomes considerable (Fig. A-1b).

However, one important caveat is that Eq. A3 and the subsequent approximation only works as long as $\epsilon_i < |1|$ is fulfilled. How likely is a violation of this criterion? Numerical simulations for $n_{\text{ref}} = 30$ appear to be very stable for any $\frac{\mu}{\sigma} > 0.8$ in the $X_{t,i}$'s, i.e. corresponding roughly to a $C_v[\hat{\mu}_{\text{ref},i}] \approx 0.2$. For such a choice of $C_v$ the chance of $|\epsilon_i| \geq 1$ corresponds to a $-5\sigma$ event with a probability of roughly 1 to 3.5 million. Given that the observed $\frac{\mu}{\sigma}$ ratios are considerably larger than the values tested here even in the driest regions of the world, we conclude that the approximation can be used for the vast majority, if not all, practical considerations.

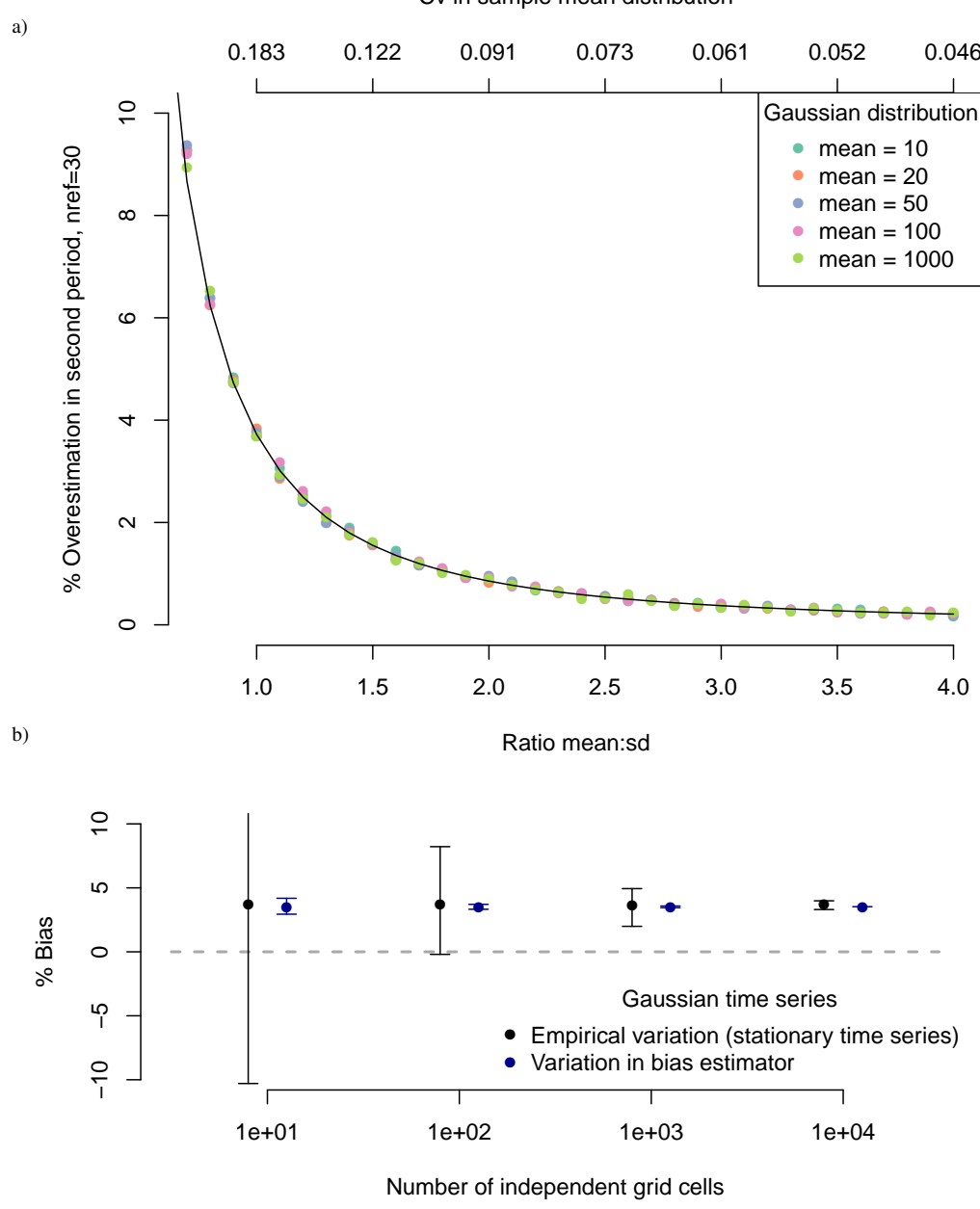

**Figure A-1.** a) Ratio of mean to sd vs. normalisation-induced bias in a Gaussian distribution for numerical simulations with various mean values (dots), and the derived analytical approximation. The reference period length is taken as $n_{ref} = 30$, and numerical simulations are conducted with $n = 10^5$ grid cells with each 60 time steps. b) Analytical estimates of biases as calculated from sample mean and sample standard deviation following Eq. 1 in the main text (dark blue) for a given number of independent grid cells ($\frac{\mu}{\sigma} = 1$, $n_{ref} = 30$). For comparison, the magnitude of random changes in stationary time series (i.e. empirical variation in the quantity $\Delta_{bias}$, following Eq. A1) with $n_{ref} = 30$ and $n_{obase} = 30$ is shown in black. Error bars indicate the 5th to 95th percentile in repeated numerical simulations.

### Generalized extreme value distribution

We investigate whether in Eq. A7 the higher-order terms in the Taylor approximation can be ignored in practical applications, where an assumption of Gaussianity might not hold. Here, we test this for the Generalized Extreme Value distribution as an appropriate model for annual maxima as investigated in the main manuscript with two different choices for the distribution's
shape parameter ($\xi$).

### i. Gumbel distribution

We first assume, in analogy to the paragraph above, independent and identically distributed (i.e., stationary) random variables drawn from a Generalized Extreme Value distribution with zero shape parameter Johnson et al. ('Gumbel distribution', $X_{t,i} \sim GEV(\mu', \sigma', \xi = 0)$, where $\mu'$, $\sigma'$ and $\xi = 0$ denote the GEV's location, scale and shape parameter, respectively 1995). The
expected values for mean ($\mu$) and variance ($\sigma^2$) of a GEV are given by $\mu = \mu' + \sigma'\gamma$, where $\gamma$ denotes Euler's constant.

Folllowing Eq. A7, we can readily derive an analytical expression for the expected value of the normalisation-induced bias, i.e.

$$\Delta_{bias} = \frac{\sigma^2}{\mu^2 n_{\text{ref}}} - \mathbf{E}(\epsilon_i^3) + \mathbf{E}(\epsilon_i^4) - \mathbf{E}(\epsilon_i^5) + ... \tag{A11}$$

$$= \left(\frac{\pi}{\sqrt{6 n_{\text{ref}}}(\frac{\mu'}{\sigma'} + \gamma)}\right)^2 - \mathbf{E}(\epsilon_i^3) + \mathbf{E}(\epsilon_i^4) - \mathbf{E}(\epsilon_i^5) + ... \tag{A12}$$

Here, we note again that the quadratic term dominates the expression. If we make the simplifying assumption that the sample means $\hat{\mu}_{\text{ref},i}$ for $n_{\text{ref}} = 30$ follow (approximately) a Gaussian distribution (the assumption is only needed for the higher order terms of the Taylor expansion), we can derive an analytical approximation for the normalisation-induced bias by insertion and in analogy to above, i.e.

$$\Delta_{bias} \approx \left(\frac{\pi}{\sqrt{6 n_{\text{ref}}}(\frac{\mu'}{\sigma'} + \gamma)}\right)^2 + \left(\frac{\sigma}{\mu} \frac{1}{\sqrt{n_{\text{ref}}}}\right)^4 \mathbf{E}(Y^4) + ... \tag{A13}$$

$$\approx \left(\frac{\pi}{\sqrt{6 n_{\text{ref}}}(\frac{\mu'}{\sigma'} + \gamma)}\right)^2 + 3\left(\frac{\pi}{\sqrt{6 n_{\text{ref}}}(\frac{\mu'}{\sigma'} + \gamma)}\right)^4. \tag{A14}$$

Hence, we find that the magnitude of the bias estimates is proportional to the ratio of scale to location parameter ($\frac{\sigma'}{\mu'}$) for any fixed reference period length (but also the proportionality to the square of the ratio of standard deviation to mean remains, i.e. Eq. 1 (or Eq. A13) in the main text). The analytical approximation can be verified by numerical simulation using GEV-distributed random variables, and is found to fit the data very well (Fig. A-2a). Furthermore, an estimator of the expected
value of the biases by only estimating the mean and standard deviation of empirical time series (i.e., using the first term in the Taylor approximation) can be derived easily and is found to work reliable also for a small number of independent grid cells (Fig. A-2c).

## ii. GEV distribution with $\xi \neq 0$

Here, we test whether the analytical argument from above can be extended to Generalized Extreme Value distributions with $\xi \neq 0$. In practical applications of the GEV to observed maximum precipitation, a shape parameter of $\xi \approx 0.2$ is often found (Van den Brink and Können, 2011), therefore we test here for $X_{t,i} \sim \text{GEV}(\mu', \sigma', \xi = 0.2)$. The expected values for mean ($\mu$) and variance ($\sigma^2$) of a GEV, when $0 > \epsilon < 1$, are given by $\mu = \mu' + \sigma' \frac{\Gamma(1-\xi)-1}{\xi}$ and $\sigma^2 = (\sigma')^2 \frac{(g_2 - g_1^2)}{\xi}$, where $g_k = \Gamma(1 - k\xi)$, $k = 1, 2$, and $\Gamma(t)$ is the gamma function (Johnson et al., 1995).

Hence, the (dominant) quadratic term in the Taylor approximation in Eq. A7 reads,

$$\Delta_{bias} \approx \frac{(g_2 - g_1^2)}{n_{\text{ref}} \xi [\frac{\mu'}{\sigma'} + \frac{\Gamma(1-\xi)-1}{\xi}]^2}. \tag{A15}$$

The approximation works again very well in numerical simulations (Fig. A-2b), and shows that if $\xi \neq 0$, there is a dependency on $\xi$, $n_{\text{ref}}$, and again the ratio of $\frac{\sigma'}{\mu'}$, which determine the magnitude of the normalisation-induced bias. Please note that the approximation works similarly well for random variables drawn from a GEV-distribution with negative shape parameter ($\xi = -0.2$, not shown).

## Short Remark on non-stationarity in the out-of-base period

Many real-world precipitation time series show non-stationarities due to climatic variations (O'Gorman, 2015) that are typically diagnosed as relative changes in the precipitation amount. Hence, we can ask whether and how any 'real change in the expected value' outside the reference period can be disentangled from the normalisation-induced bias. Given the analytical approximation above, we can show that the highlighted normalisation-induced bias scales non-stationarities in the out-of-base period in a multiplicative way:

Let $c$ denote any change between the reference period expected value and some future period (e.g. assume one is interested in global or latitudinal changes in a past and future climatic period), i.e. such that $\mathbf{E}(X_{t_{\text{ref}},i}) = c\mathbf{E}(X_{t_{\text{oob}},i})$, then the bias ($\Delta_{\text{bias}}$, after accounting for the 'real change') would simply scale with the relative change ($\Delta$ denotes the total apparent change):

$$\Delta = c\mathbf{E}\left(\frac{X_{t,i}}{\hat{\mu}_{\text{ref},i}}\right) - 1 \tag{A16}$$

$$= c\mathbf{E}\left(\frac{1}{1+\epsilon_i}\right) - 1 \tag{A17}$$

$$= \underbrace{c-1}_{\text{true change}} + c[\underbrace{\frac{\sigma^2}{\mu^2 n_{\text{ref}}} - \mathbf{E}(\epsilon_i^3) + \mathbf{E}(\epsilon_i^4) - \mathbf{E}(\epsilon_i^5) + ...]}_{\Delta_{\text{bias}}} \tag{A18}$$

From Eq. A18, it is straightforward to see that for any multiplicative changes in the expected value of the out-of-base variables, the normalisation-induced bias scales with the change. Hence, this expression implies that to detect the 'true change $c$' between two periods, the normalisation-induced bias has to be accounted for, i.e.

$$c = \frac{\Delta + 1}{1 + \Delta_{\text{bias}}}. \tag{A19}$$

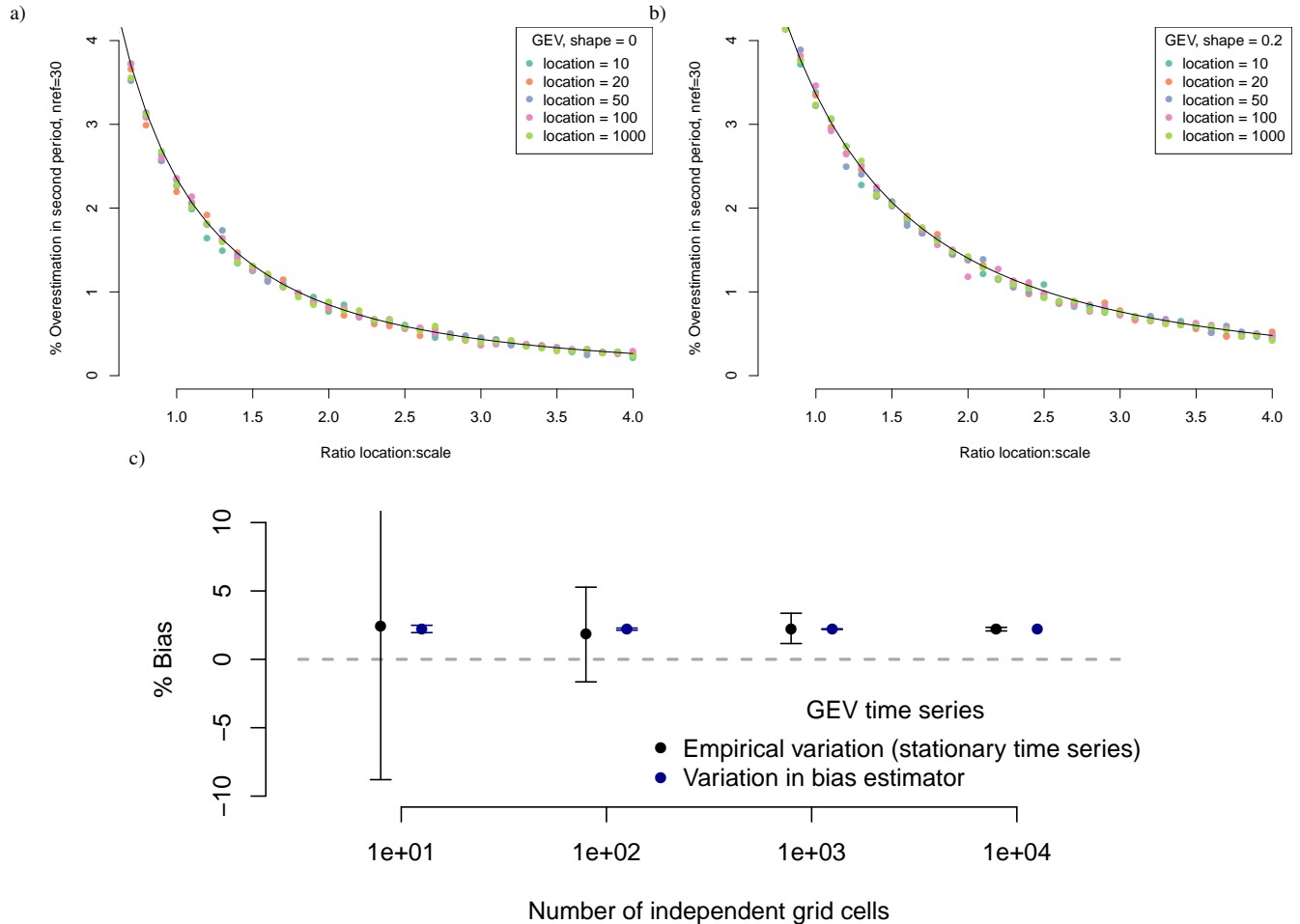

**Figure A-2.** a) Ratio of location to scale parameter vs. normalisation-induced bias in a Generalized extreme value distribution for numerical simulations with various location parameter values (dots) and a) zero shape parameter, and b) with $\xi = 0.2$. Reference period length is taken as $n_{\text{ref}} = 30$, and numerical simulations are conducted with $n = 10^5$ grid cells with each 60 time steps. c) Analytical estimates of biases as calculated from sample mean and sample standard deviation following Eq. 1 in the main text (dark blue) for a given number of independent grid cells drawn from a GEV distribution ($\frac{\mu'}{\sigma'} = 1$, $\xi = 0$, $n_{\text{ref}} = 30$). For comparison, the magnitude of random changes in stationary time series (i.e. empirical variation in the quantity $\Delta_{bias}$, following Eq. A1) with $n_{\text{ref}} = 30$ and $n_{\text{obase}} = 30$ is shown in black. Error bars indicate the 5th to 95th percentile in repeated numerical simulations.

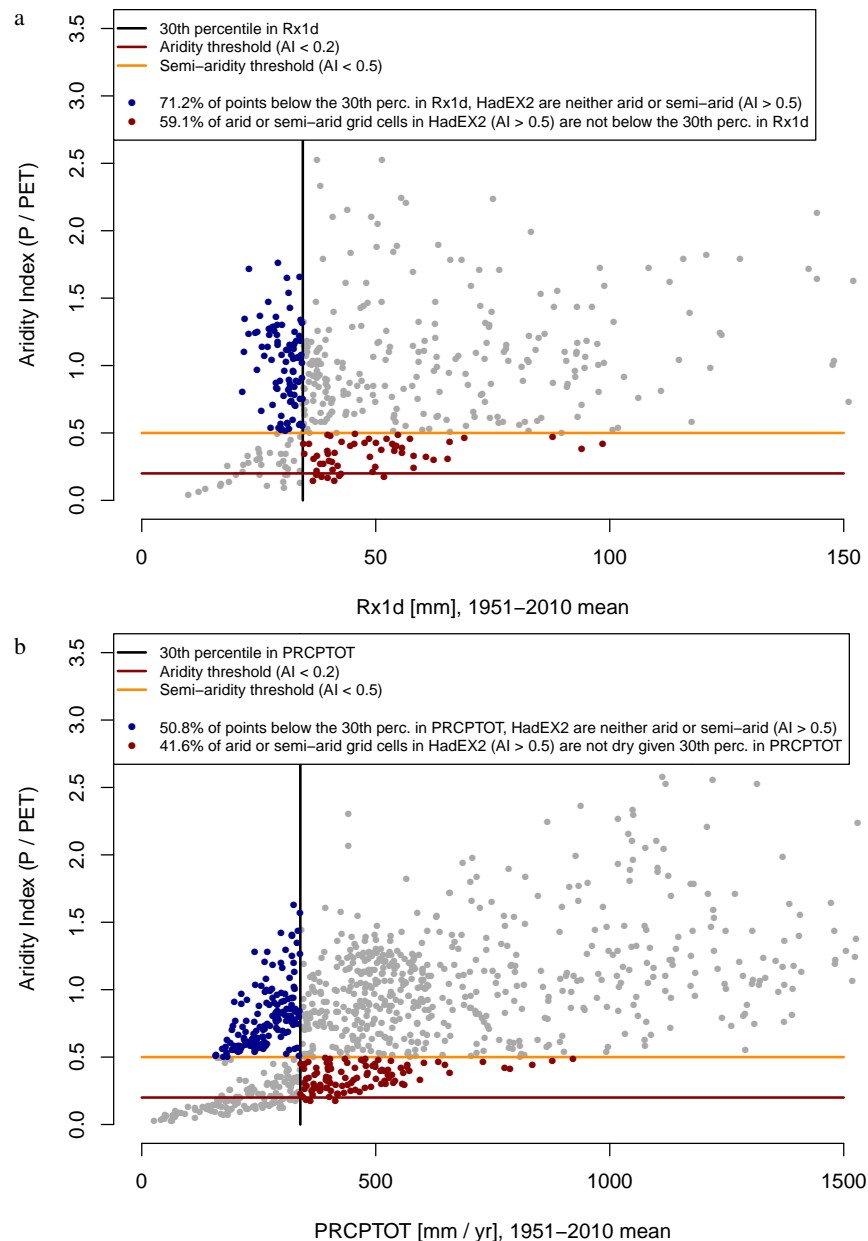

**Figure B-1.** Relationship between annual-maximum daily rainfall (Rx1d from HadEX2-GHCNDEX merged dataset) and aridity (a), and precipitation totals (PRCPTOT from HadEX2-GHCNDEX merged dataset) and aridity (b). Potential evapotranspiration is taken from the CRU-TS3.23 dataset (Harris et al., 2014)

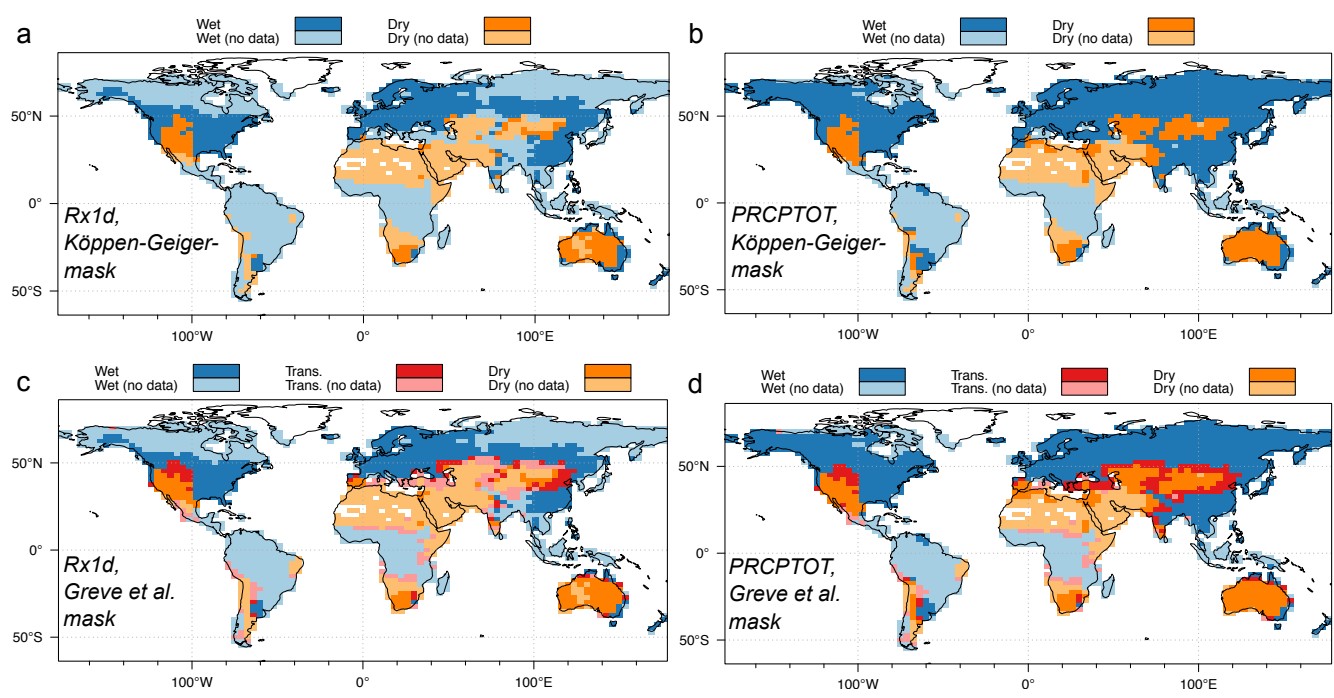

**Figure B-2.** Available data in the HadEX2 dataset (Donat et al., 2013a) merged with GHCNDEX (Donat et al., 2013b).

*Author contributions.* S.S. and J.Z. conceived the study. All authors contributed to writing the paper.

*Acknowledgements.* S.S. and M.D.M. are grateful to the European Commission for funding the BACI project (grant agreement No 640176) and to the European Space agency for funding the STSE project CAB-LAB. We thank M. Donat for discussions and four anonymous reviewers for comments on an earlier form of the manuscript. Discussions with Holger Metzler and Fabian Gans on the analytical approximation presented in the Appendix are gratefully acknowledged.

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

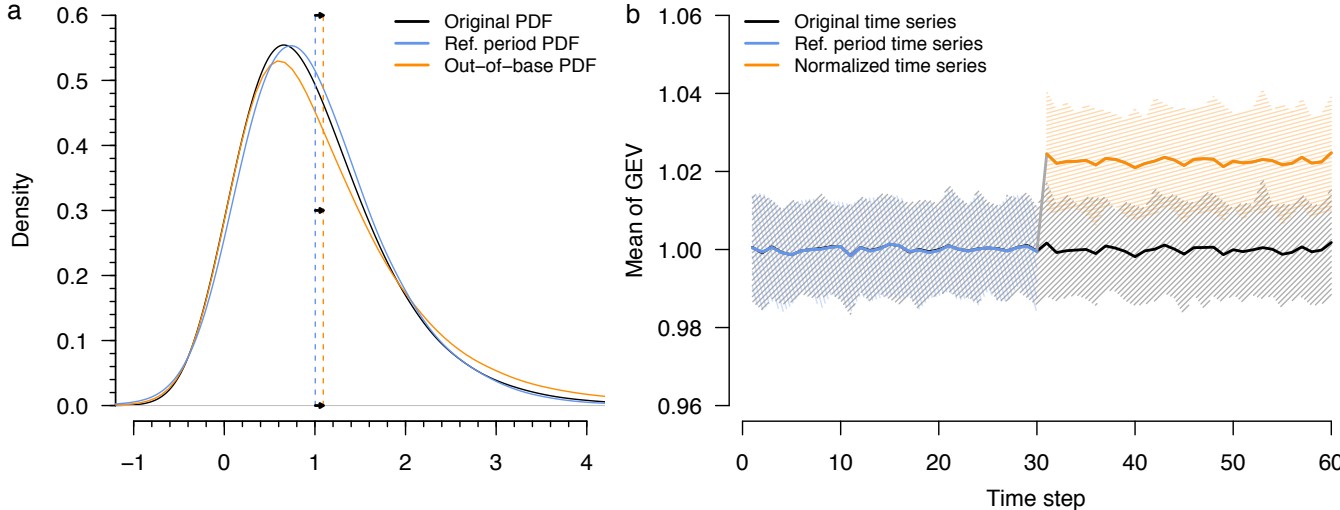

**Figure 1.** Conceptual example of biases in the mean induced by normalisation based on a fixed reference period. a, Probability distributions and their respective means for an artificial dataset of $10^4$ grid cells each comprised of random variables sampled from a Generalized Extreme Value distribution (GEV, $\mu = 1$, $\sigma = 1$, $\xi = 0$, sample size $n_{ref} = 8$ for illustration), and normalised following Donat et al. (2016) with different ref. periods. b, Shift in the mean of spatially aggregated variables due to reference period normalisation ($n_{ref} = 30$ following Donat et al. (2016), Confidence intervals denote the 5th - 95th percentile). Code to reproduce this example is provided in Supplementary Material.

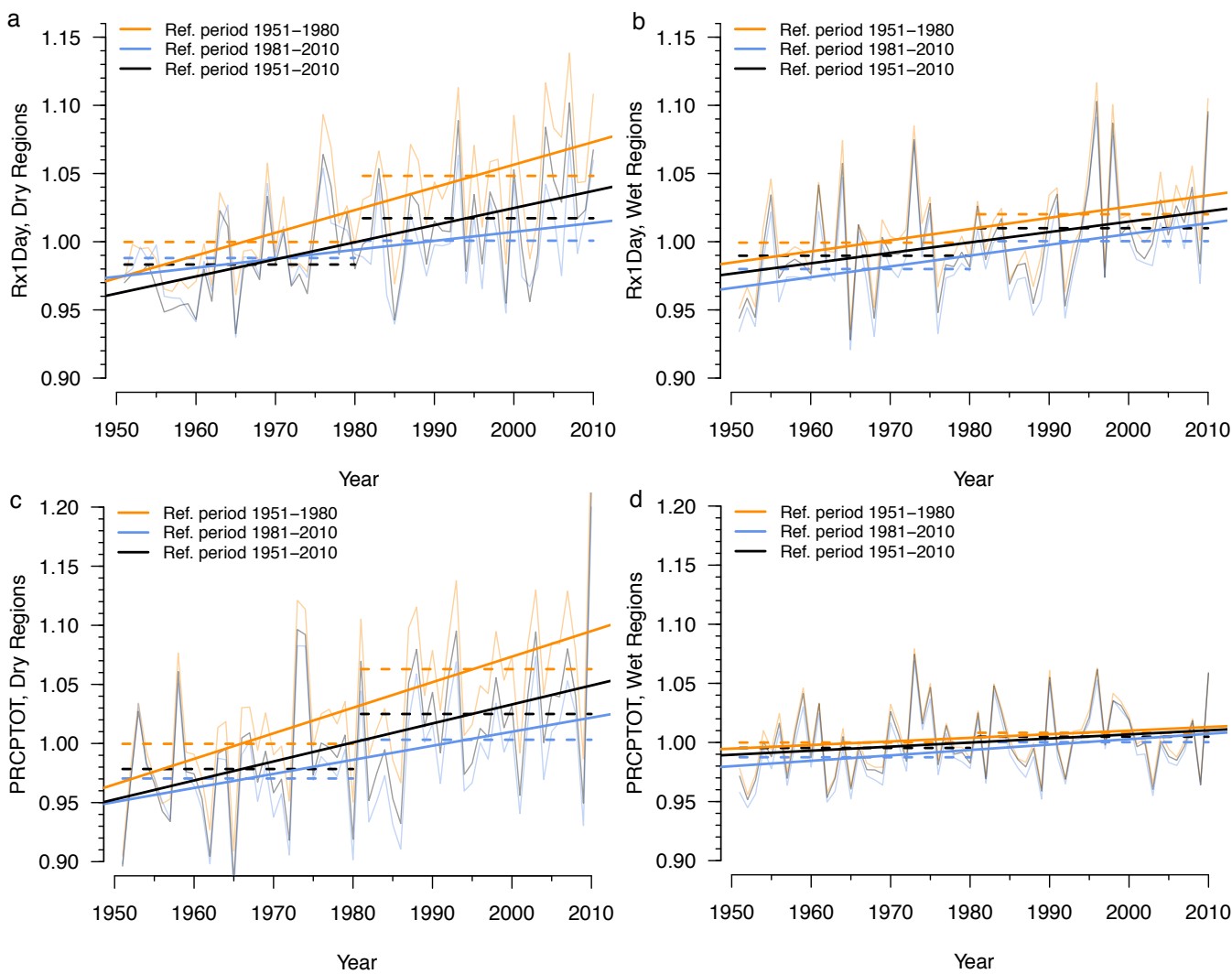

**Figure 2.** Normalisation-induced biases on time series and trend estimates. a-b, Time series, trends and 30-year means of spatially aggregated heavy precipitation (Rx1d) in (a) dry and (b) wet regions. c-d, Time series, trends and 30-year means of spatially aggregated total precipitation (PRCPTOT) in (a) dry and (b) wet regions. Orange lines are taken from Donat et al. (2016) (ref. period: 1951-1980), black lines are corrected for biases (ref. period: 1951-2010), and blue lines indicate a hypothetical 1981-2010 reference period.

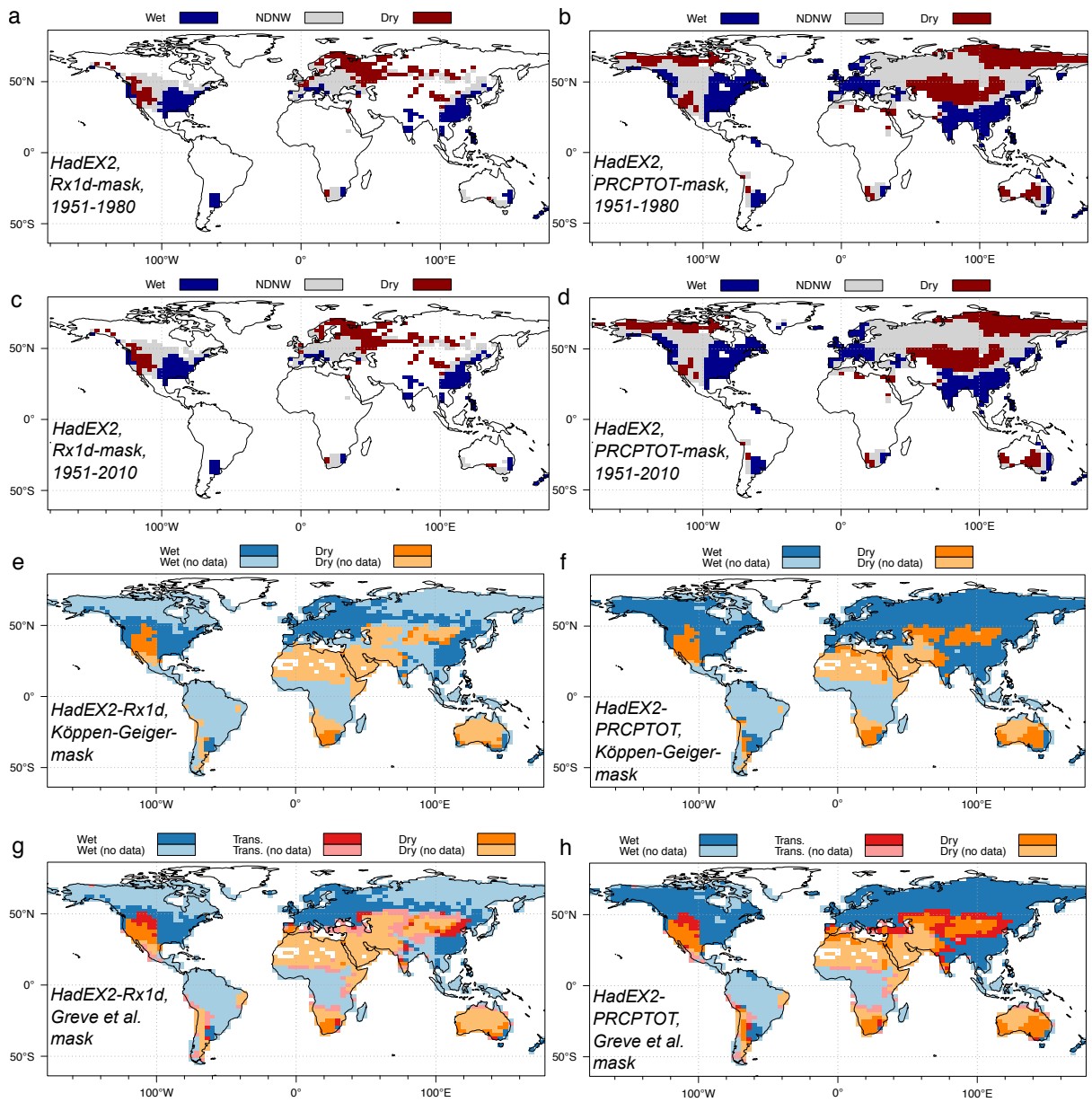

**Figure 3.** Different mask of the world's dry and wet regions. a-d, Dryness/Wetness masks based on 1951-1980 and HadEX2 (a-b, (see Donat et al., 2016)) and 1951-2010 (c-d, to avoid 'regression to the mean' selection bias, see main text) for Rx1d (left) and PRCPTOT (right). 'NDNW' indicates neither dry nor wet areas, white inland areas indicate less than 90% data availability in the HadEX2 dataset and were not considered. e-f, Dry regions based on the Köppen-Geiger classification as updated by (Kottek et al., 2006) and data availability in HadEX2. g-h, Dry and transitional regions following (Greve et al., 2014) and data availability in HadEX2.

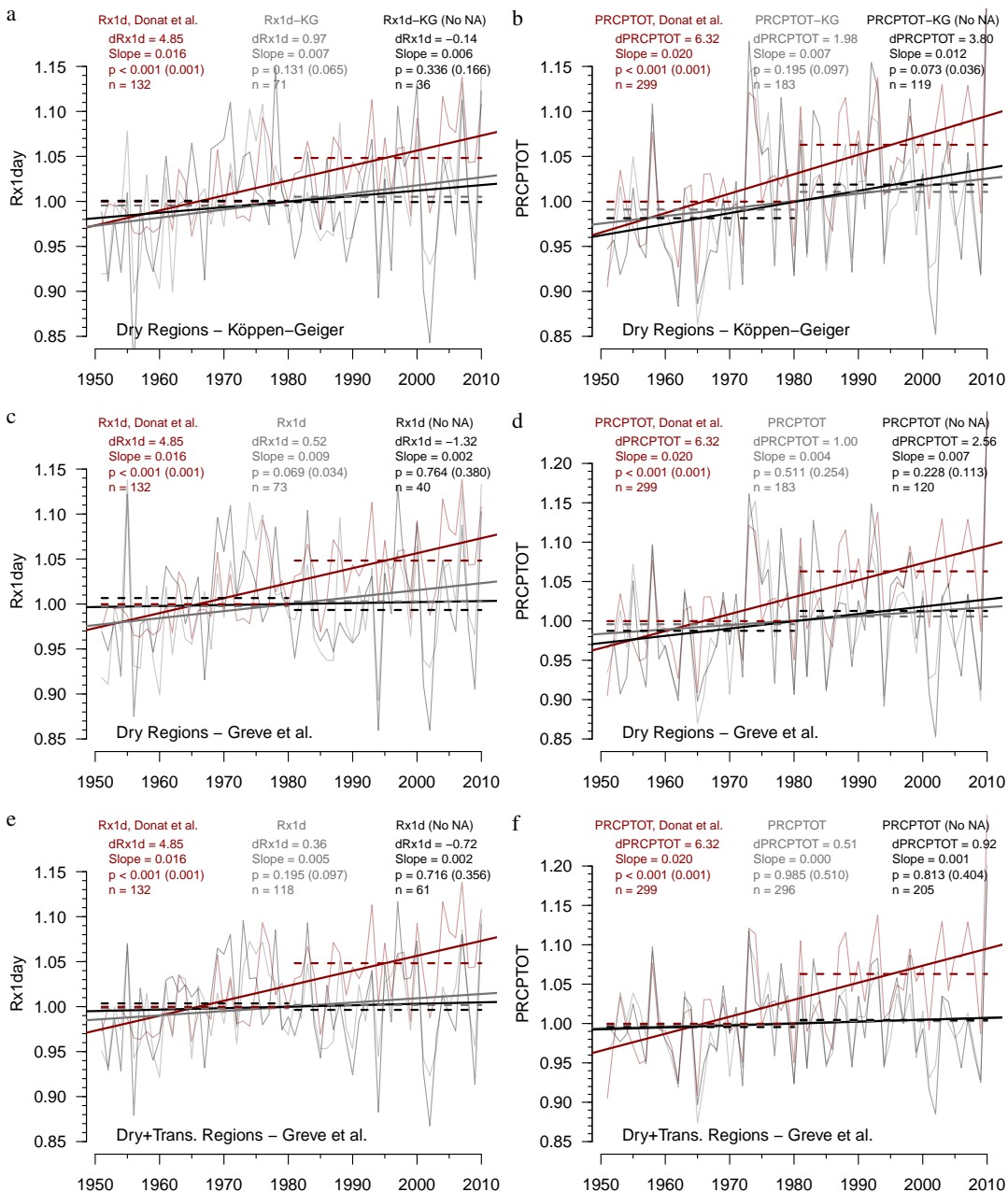

**Figure 4.** a-f, Time series, trends and 30-year means of spatially aggregated heavy precipitation (Rx1d, a,c,e) and annual rainfall totals (PRCPTOT, b,d,f) in dry regions following (a-b) the Köppen-Geiger classification Kottek et al. (2006), (c-d) Greve et al. (2014), and (e-f) dry and transitional regions combined (Greve et al., 2014). Red lines are drawn as reported in Donat et al. (2016) for comparison, i.e. based on the 1951-1980 reference period and dryness defined as 'moderate extreme precipitation' (Rx1d) and annual precipitation totals (PRCPTOT). Grey and black lines are corrected for statistical artefacts (1951-2010 reference period), and dry regions are defined based on aridity. Grey lines report 90% complete time series, black lines report only data with 100% complete temporal coverage. All p-values are given for two-sided (one-sided) Mann-Kendall trend tests.

**Table 1.** Statistical pre-processing uncertainties and biases in period increments and trend slopes

| World Region | Precipitation characteristic | Ref. Period (Normalisation) | Ref. Period (region selection) | Period Increment[1] [%] | Bias [%] | Sen slope [decade$^{-1}$] | Bias [%] | Type of bias |
|---|---|---|---|---|---|---|---|---|
| | Rx1d | 1951-1980 | 1951-1980 | 4.85 | 40.4 | 0.016 | 33.3 | [2] |
| Dry (HadEX2, | Rx1d | 1981-2010 | 1981-2010 | 1.29 | -62.7 | 0.006 | -50.0 | [3] |
| 30% lowest | Rx1d | 1951-2010 | 1951-2010 | 3.45 | 0.0 | 0.012 | 0.0 | [4] |
| Rx1day) | Rx1d | 1951-1980 | 1951-2010 | 3.97 | 15.1 | 0.014 | 16.7 | [5] |
| | Rx1d | 1951-2010 | 1951-1980 | 4.33 | 25.3 | 0.014 | 16.7 | [6] |
| | Rx1d | 1951-1980 | 1951-1980 | 2.09 | 2.2 | 0.007 | 8.7 | [2] |
| Wet (HadEX2, | Rx1d | 1981-2010 | 1981-2010 | 2.09 | 2.2 | 0.007 | -1.5 | [3] |
| 70% highest | Rx1d | 1951-2010 | 1951-2010 | 2.04 | 0.0 | 0.007 | 0.0 | [4] |
| Rx1day) | Rx1d | 1951-1980 | 1951-2010 | 2.41 | 18.1 | 0.008 | 16.0 | [5] |
| | Rx1d | 1951-2010 | 1951-1980 | 1.73 | -15.3 | 0.006 | -4.8 | [6] |
| | PRCPTOT | 1951-1980 | 1951-1980 | 6.32 | 32.9 | 0.020 | 40.4 | [2] |
| Dry (HadEX2, | PRCPTOT | 1981-2010 | 1981-2010 | 3.38 | -29.0 | 0.010 | -29.5 | [3] |
| 30% lowest | PRCPTOT | 1951-2010 | 1951-2010 | 4.76 | 0.0 | 0.015 | 0.0 | [4] |
| PRCPTOT) | PRCPTOT | 1951-1980 | 1951-2010 | 5.74 | 20.8 | 0.019 | 27.5 | [5] |
| | PRCPTOT | 1951-2010 | 1951-1980 | 5.34 | 12.2 | 0.017 | 14.9 | [6] |
| | PRCPTOT | 1951-1980 | 1951-1980 | 0.83 | -13.7 | 0.003 | -13.6 | [2] |
| Wet (HadEX2, | PRCPTOT | 1981-2010 | 1981-2010 | 1.30 | 35.5 | 0.005 | 28.9 | [3] |
| 70% highest | PRCPTOT | 1951-2010 | 1951-2010 | 0.96 | 0.0 | 0.004 | 0.0 | [4] |
| PRCPTOT) | PRCPTOT | 1951-1980 | 1951-2010 | 1.32 | 38.5 | 0.005 | 38.2 | [5] |
| | PRCPTOT | 1951-2010 | 1951-1980 | 0.40 | -58.6 | 0.002 | -52.4 | [6] |

[1] Period increment denotes the change in period means between 1981-2010 vs. 1951-1980.

[2] Combination of 'Normalisation' and 'Regression to mean' (RTM) bias, 'early' ref. period (i.e. following Donat et al. (2016))

[3] Combination of 'Normalisation' and 'RTM' bias, 'late' ref. period

[4] Ref. Period covering the entire temporal domain (no bias)

[5] 'Normalisation' bias only

[6] 'RTM' bias only

**Table 2.** Uncertainties regarding the definition of a 'dry region', Rx1d.

| Dry Region Definition | Dataset | Ref. Period | Temporal Coverage (%) | Period Increment[1] [%] | Trend Slope [decade$^{-1}$] | two-sided (one-sided) p-value | Sample size |
|---|---|---|---|---|---|---|---|
| Donat et al. (2016), global 30% quantile in Rx1d | HadEX2[2] | 1951-1980 | 90% | 4.85 | 0.016 | <0.001 (<0.001) | 132 |
| Donat et al. (2016), global 30% quantile in Rx1d | HadEX2[2] | 1951-1980 | 100% | 6.07 | 0.020 | <0.001 (<0.001) | 57 |
| Donat et al. (2016), global 30% quantile in Rx1d | HadEX2[2] | 1951-2010 | 90% | 3.45 | 0.012 | <0.001 (<0.001) | 132 |
| Donat et al. (2016), global 30% quantile in Rx1d | HadEX2[2] | 1951-2010 | 100% | 4.24 | 0.017 | <0.001 (<0.001) | 57 |
| Köppen (1900), dry climates ('B-climates') | HadEX2[2] | 1951-2010 | 90% | 0.97 | 0.007 | 0.131 (0.064) | 71 |
| Köppen (1900), dry climates ('B-climates') | HadEX2[2] | 1951-2010 | 100% | −0.14 | 0.006 | 0.336 (0.167) | 36 |
| Greve et al. (2014), dry regions | HadEX2[2] | 1951-2010 | 90% | 0.52 | 0.009 | 0.069 (0.034) | 73 |
| Greve et al. (2014), dry regions | HadEX2[2] | 1951-2010 | 100% | −1.32 | 0.002 | 0.764 (0.380) | 40 |
| Greve et al. (2014), dry+transitional regions | HadEX2[2] | 1951-2010 | 90% | 0.36 | 0.005 | 0.195 (0.097) | 118 |
| Greve et al. (2014), dry+transitional regions | HadEX2[2] | 1951-2010 | 100% | −0.72 | 0.002 | 0.716 (0.356) | 61 |
| Köppen (1900), dry climates ('B-climates') | HadEX2-GHCNDEX[3] | 1951-2010 | 90% | 1.41 | 0.011 | 0.058 (0.029) | 127 |
| Köppen (1900), dry climates ('B-climates') | HadEX2-GHCNDEX[3] | 1951-2010 | 100% | 0.68 | 0.008 | 0.101 (0.050) | 78 |
| Greve et al. (2014), dry regions | HadEX2-GHCNDEX[3] | 1951-2010 | 90% | 1.16 | 0.010 | 0.049 (0.024) | 124 |
| Greve et al. (2014), dry regions | HadEX2-GHCNDEX[3] | 1951-2010 | 100% | 0.00 | 0.005 | 0.243 (0.121) | 80 |
| Greve et al. (2014), dry+transitional regions | HadEX2-GHCNDEX[3] | 1951-2010 | 90% | 0.81 | 0.007 | 0.099 (0.049) | 191 |
| Greve et al. (2014), dry+transitional regions | HadEX2-GHCNDEX[3] | 1951-2010 | 100% | −0.16 | 0.004 | 0.270 (0.134) | 120 |

[1] Period increment denotes the change in period means between 1981-2010 vs. 1951-1980.

[2] HadEX2 is the same dataset used in the original study (Donat et al., 2016).

[3] HadEX2-GHCNDEX is a merged version, where GHCNDEX data (Donat et al., 2013b) has been added to HadEX2 data in arid regions.

**Table 3.** Uncertainties regarding the definition of a 'dry region', PRCPTOT.

| Dry Region Definition | Dataset | Ref. Period | Temporal Coverage (%) | Period Increment[1] (%) | Trend Slope (decade$^{-1}$) | two-sided p-value (one-sided) | Sample size |
|---|---|---|---|---|---|---|---|
| Donat et al. (2016), global 30% quantile in PRCPTOT | HadEX2[2] | 1951-1980 | 90% | 6.32 | 0.020 | < 0.001 (< 0.001) | 299 |
| Donat et al. (2016), global 30% quantile in PRCPTOT | HadEX2[2] | 1951-1980 | 100% | 5.93 | 0.015 | 0.002 (0.001) | 108 |
| Donat et al. (2016), global 30% quantile in PRCPTOT | HadEX2[2] | 1951-2010 | 90% | 4.76 | 0.015 | < 0.001 (< 0.001) | 299 |
| Donat et al. (2016), global 30% quantile in PRCPTOT | HadEX2[2] | 1951-2010 | 100% | 4.37 | 0.010 | 0.157 (0.077) | 108 |
| Köppen (1900), dry climates ('B-climates') | HadEX2[2] | 1951-2010 | 90% | 1.98 | 0.007 | 0.195 (0.100) | 183 |
| Köppen (1900), dry climates ('B-climates') | HadEX2[2] | 1951-2010 | 100% | 3.80 | 0.012 | 0.073 (0.036) | 119 |
| Greve et al. (2014), dry regions | HadEX2[2] | 1951-2010 | 90% | 1.00 | 0.004 | 0.511 (0.254) | 183 |
| Greve et al. (2014), dry regions | HadEX2[2] | 1951-2010 | 100% | 2.56 | 0.007 | 0.228 (0.113) | 120 |
| Greve et al. (2014), dry+transitional regions | HadEX2[2] | 1951-2010 | 90% | 0.51 | 0.000 | 0.985 (0.510) | 296 |
| Greve et al. (2014), dry+transitional regions | HadEX2[2] | 1951-2010 | 100% | 0.92 | 0.001 | 0.813 (0.404) | 205 |
| Köppen (1900), dry climates ('B-climates') | HadEX2-GHCNDEX[3] | 1951-2010 | 90% | 3.47 | 0.013 | 0.030 (0.015) | 234 |
| Köppen (1900), dry climates ('B-climates') | HadEX2-GHCNDEX[3] | 1951-2010 | 100% | 5.14 | 0.019 | 0.009 (0.004) | 175 |
| Greve et al. (2014), dry regions | HadEX2-GHCNDEX[3] | 1951-2010 | 90% | 2.63 | 0.011 | 0.077 (0.038) | 231 |
| Greve et al. (2014), dry regions | HadEX2-GHCNDEX[3] | 1951-2010 | 100% | 4.20 | 0.017 | 0.024 (0.012) | 170 |
| Greve et al. (2014), dry+transitional regions | HadEX2-GHCNDEX[3] | 1951-2010 | 90% | 1.67 | 0.006 | 0.200 (0.099) | 356 |
| Greve et al. (2014), dry+transitional regions | HadEX2-GHCNDEX[3] | 1951-2010 | 100% | 2.47 | 0.009 | 0.084 (0.041) | 275 |

[1] Period increment denotes the change in period means between 1981-2010 vs. 1951-1980.

[2] HadEX2 is the same dataset used in the original study (Donat et al., 2016).

[3] HadEX2-GHCNDEX is a merged version, where GHCNDEX data (Donat et al., 2013b) has been added to HadEX2 data in arid regions.