# Peer review of "Have precipitation extremes and annual totals been increasing in the world's dry regions over the last 60 years?"

_Hydrology and Earth System Sciences, 2016_

## Referee Comment (RC1) · Anonymous Referee #1 · 11 Oct 2016

This manuscript highlights two concerns with the recent analysis presented in Donat et al (2016), who reported increasing trends in extreme precipitation and total precipitation in dry regions around the world. The two concerns raised by the authors with the analysis of Donat et al (2016) are valid. Here the authors present a re-analysis of the Donat et al (2016) work using more appropriate methodologies and find the results of Donat et al (2016) are highly dependent on the methodology they adopted, which has significant implications for the conclusions of Donat et al (2016). Overall, this manuscript presents an important critique of Donat et al (2016), which is highly relevant to the general scientific community. I believe the manuscript will be worthy of publication following moderate revision to improve the clarity of the manuscript, as it is

hard to follow at times.

My key comments / suggestions are as follows.

The overall style of this manuscript is abbreviated and densely packed. In fact, some sections are difficult to follow as helpful explanations are not provided. Figures and Tables are cited, but little accompanying explanation is provided to help the reader understand and interpret them. The current manuscript style is like a very compact 'communication arising'. I think this style provides the key messages, but it is very difficult to follow the technical argument in places. Also, why are Appendix A and B not just normal Figures and Tables, like the other Figures and Tables? Why the separate Appendices? I recommend you move this material from the Appendices into the paper.

The key points made in this manuscript are fine, but the explanation accompanying the Figures and Tables requires expansion to improve the readability of the manuscript. At times I found it difficult to know how each Figure and Table slotted into the overall story; not because the material is irrelevant, but because a context for the material was not provided. There is a lot of interesting material in the Figures and Tables, which is hardly covered in the text. Expanding the explanations around the Figures and Tables will guide the reader through this important material and significantly increase the accessibility of this research.

Minor comments

Page 3, line 18: "the dataset will result". Are you 100% certain it will result in a higher probability for wetter (drier) conditions. Or is a better word to use here "may" result. I think this paragraph would benefit from an expanded explanation of the statistical bias being discussed as it is not easy to follow.

Appendix B second Table: Replace "Rx1day" with "PRCPTOT" in the wet and dry regions.

Figure 1: you need to improve the explanation of this Figure. The illustration provided

[Figure]

in the text (page 3, lines 6-13) was excellent, but the connection to Figure 1 was not obvious.

Figure 3c, 3d: are the p-values for the one-sided and two-sided trend tests reported correct or have they been switched?

Tables 1 & 2: Explain what Period Inc. means.

---

## Referee Comment (RC2) · Anonymous Referee #2 · 20 Oct 2016

Review of Sippel et al., 'Have precipitation extremes and annual totals been increasing in the world's dry regions over the last 60 years?'

This paper (which can effectively be considered as a comment on the Donat et al (2016) paper) raises two issues with the Donat et al (2016) (hereafter D2016) paper – the way in which spatial averaging has been used and the way in which dry regions have been defined.

Both of these are legitimate concerns. However, in my view both this paper and D2016 miss what I think is the main point with respect to the definition of dry regions – namely, that most of the world's driest regions (in particular, almost all of the Sahara and the Middle East) are excluded because of a lack of data. (Similarly, many of the world's

wetter regions in South America, equatorial Africa and southeast Asia are also excluded). Any definition, whether it is the one used in D2016 or in this paper, is likely to give an unrepresentative sample of the world's dry regions given that the data availability is largely confined to North America, Eurasia and parts of Australia. Put another way, the HadEX2 data set in its current form is not capable of providing a fully representative sample of the world's dry regions, which is particularly important given that there is no reason why we would expect tropical arid and semi-arid zones (e.g. the Sahel), subtropical deserts (e.g. southwest US) and high-latitude low-precipitation regions to have similar long-term trends. A casual reader encountering either this paper or D2016 would expect the papers to be covering a very different range of areas to that which they actually do.

(I would view both this paper's method and the D2016 method as being reasonable ways of defining dry regions – the issue is that neither is representative given the gaps in the data set).

Averaging precipitation indices is another challenge – whilst the averaging period (as mentioned in this paper) is one issue, another is whether it is appropriate to average values from a distribution which is bounded below by zero and highly non-Gaussian. If one averages absolute values, area averages are likely to be dominated by the wetter areas; if one averages normalised values, there will be much more volatility in the driest areas. (Somewhat ironically, the fact that the HadEX2 data set excludes most of the world's really dry areas averted a bigger problem here – in climates where mean annual values are, say, below 10 mm, annual totals in excess of 1000% are plausible, which would completely overwhelm less variable climates in a spatial average). In my view, it would be better not to try to do spatial averages at all, and instead report using indicators such as the % of gridpoints which show significant positive/negative trends. That said, if you are going to average precipitation indices, then this paper has identified a genuine issue with the D2016 methodology.

In summary – I think this paper accurately documents valid issues with the D2016

paper, and as such I think it is appropriate for publication, but I also think it would be improved if it engages substantially more with the issues identified above.

---

## Referee Comment (RC3) · D. Stone (Referee) · 21 Oct 2016

The authors examine the robustness to choices made in the analysis of a recent analysis of observed trends in precipitation in dry regions around the world. In general I quite like this, as results of studies are usually interpreted beyond the specific experiment design of the analysis, and so this paper performs the important task of determining the extent to which it is possible in the case of observed precipitation trends in dry regions. However, I think there are a couple of additional aspects to this that the authors have not considered, as well as one important syntactic issue, that I believe need to be addressed before publication.

First, the motivation you frequently mention is for informing adaption decisions. For

that motivation, though, it is not clear to me that what you do in terms of normalising to the full period is necessarily any more appropriate than what Donat et alii (2016) did. Many decisions are based on climatic or hydrologic data from a specific time period, for instance in the case of international treaties allocating water on an international river. Thus adaptation decisions need to be made with respect to divergence from that reference baseline (ignoring non-climate stuff). So while e.g. you may be correct that there has been no actual trend in precipitation totals, say, that does not necessarily mean there has not been a trend as measured by stipulated monitoring procedures used in many decision-making settings. Cast another way, we have the same problem in dealing with future climate change: projections are based on, say, the full historical period you use, but that does not include the future itself. I expect you are not arguing that we cannot make useful projections of future climate change simply because we have not monitored the future yet. In this context, I laud your effort because you highlight the sensitivity to this point, but I think it is important – and entirely consistent with you consideration of robustness – to emphasise that there is not necessarily a single "correct" answer.

Second, in terms of all of the discussion about what constitutes a "dry" region, the most striking thing to me is that none of the definitions of dry regions you consider include what I think of as prototypical dry regions: the Sahara, the Saudi Peninsula, Central Asia (particulaly for Rx1D), southwestern Africa (other than South Africa), western Australia, northern Mexico (for Rx1D), nor the driest areas of South America (for Rx1D). The reason for this of course is monitoring coverage, but given the absence of all of these regions (the Sahara!) I do not think these results can plausibly be considered as being indicative of how precipitation is actually changing over the world's dry regions. Again, I consider this a point about robustness that is entirely consistent with your paper, but it most certainly needs to be acknowledged/noted/highlighted.

Third, on the syntactic side, while the title refers to precipitation and it appears to be precipitation you are actually analysing, within the text this is generally referred to as

"rainfall". Please clarify which you are examining, as these are certainly not identical for annual totals (and, if defined in certain ways, for heavy extremes) in many of the regions you examine.

Specific comments:

page 1, line 1 The title says you are examing precipitation extremes and annual totals, but here you indicate it is rainfall. Which is it? It seems to be a precipitation dataset you are using, so it looks like the usage of "rainfall" is wrong?

page 2, lines 24-25 If they are being underestimated, then it sounds like the errors are not completely cancelling, right?

page 2, line 25 "These results": Which, Donat's or yours?

page 2, lines 25-26 I think such an assertive statement concerning the decision-making processes utilised in dry regions requires some supporting evidence, e.g. to other research on decision-making in those regions.

page 3, line 9 "in both time periods" -> "over the combined periods"

page 4, lines 21-22 This is a case where if you are considering rainfall, and not precipitation, then indeed North-East Siberia is rather dry.

page 4, lines 25-26 I do not believe that Fischer and Knutti (2015) studied the decision-making processes used by those involved in responding to climate change, and in particular what they considered "relevant" information for informing those processes.

page 6 "Figure 0" should have a different identifier.

page 6, caption Can you confirm that for only different between columns for the lowest two rows is the mask? I found this caption confusing, for instance with the distinction between the columns being introduced only halfway through. Subtitles on each panel could help.

Figure 2, caption line 3 By "red lines" do you mean yellow?

Tables 1 and 2 What does "Period Inc. (%)" mean?

Tables 1 and 2 Why do the "Sample size" values differ? Aren't all the trends calculated over the same number of years?

Table 2 There is one trend values listed as "<0.000". Why do you not give the numerical value for a negative trend? This one is interesting, because it is the only significant negative trend.

Sincerely, Dáithí Stone

————————————————————

---

## Short Comment (SC1) · 24 Oct 2016

The authors scrutinize a recent study (Donat et al. 2016) that reported increasing trends in precipitation extremes and annual totals in the world's dry regions, as defined by precipitation amounts. The authors (1) suggest that the results of the scrutinised study were biased owing to choices of the reference period, and (2) discuss that the findings depend on how 'dry' regions are defined.

We thank the authors for pointing out the statistical issue related to the reference period which is now addressed in a Corrigendum (submitted to Nature Climate Change on 12th September 2016). Importantly, this statistical issue does not affect

the major conclusions of the scrutinised study, a point that should be made clearly in the current manuscript. However, the remainder of this manuscript, in particular the discussion related to the definition of dry regions, is biased, inconsistent, ambiguous (misleading), and incomplete as outlined below. Therefore the manuscript needs to be carefully revised before publication.

***Biased:*** The current manuscript claims that the only valid definitions of wet and dry regions are those based on surface water availability, referring to what is 'commonly understood' or 'conventional'. However, in everyday language it is common to use 'wet' or 'dry' to refer to high or low precipitation for both regions and times of year. Furthermore, in the scientific literature there are numerous related studies that have defined wet and dry solely based on meteorological parameters such as precipitation (e.g. Allan et al.,2010; Sun et al., 2012; Liu and Allan, 2013), and these are ignored in the current discussion and should be included in a revised manuscript. The current manuscript, therefore, appears biased in that it is largely based on a claim that only a particular definition of dryness is valid, when several other definitions are in common use.

An important point that emerges from this discussion is that it is desirable to specify more clearly which type of definition of dry and wet is being used in studies of climate change. Indeed this is something the current manuscript could do better; see 'ambiguous' section below. We suggest to the authors that they make the conclusion of their paper and abstract a call for more specificity in the use of 'dry' and 'wet' in climate-change studies. For example, one could refer to 'meteorological' or 'hydrological' wet and dry regions, by analogy with the standard definitions of 'meteorological' or 'hydrological' drought. This would be of greater value than arguing that only one type of definition is valid.

***Inconsistent:*** The analysis in Section 3 is likely affected by the same "regression to the mean" bias discussed in Section 2, because the dry-regions masks that include water demand were not defined over the entire study period 1951-2010.

***Ambiguous:*** The current text uses 'dry' for different concepts, and this is likely to confuse readers. To avoid confusion, the authors should specify whether they are talking about 'low-precipitation' or 'arid'/'water-limited' regions. This is particularly problematic e.g. in the Abstract lines 3-5 where dry is defined in terms of water availability but then immediately used to refer to the scrutinised study in which dry means low precipitation. Similarly in the introduction it needs to be specified which concepts of 'dry' the authors refer to in each case.

***Incomplete:*** The main reason why Sippel et al. don't find a (statistically significant) increase in Rx1day in arid regions seems to be related to scarcity of data. It is unfortunate reality that arid regions are insufficiently covered by observations. Aggregating only over a few grid cells results in relatively noisy time series, so that – despite a positive trend slope – the p-value of the applied trend test is too high to reject the null hypothesis of 'no change'. A relatively easy attempt to optimise spatial coverage by merging the two existing datasets HadEX2 (Donat et al., 2013a) and GHCNDEX (Donat et al., 2013b) gives a few additional grid cells with data in arid regions. Aggregating over this just slightly improved coverage results in a more robust trend estimate in observations and in the CMIP5 ensemble mean (Figure 1). This suggests that a major uncertainty when analysing precipitation changes in arid regions comes from the limited availability of observations. Also, if using the complete coverage as provided e.g. by the ensemble of CMIP5 models as used in Donat et al. (2016), the authors would find statistically significant increases in ensemble mean over the arid regions (not shown). Therefore we assume that the main reason why Sippel et al. conclude there is 'no significant increase in heavy precipitation' in arid regions is related to the scarcity of observations.

**Specific comments:**

Page 2, line 3: 'if there is enough moisture available' – do you mean annual average moisture availability? Or seasonal? Or on the day the rainfall extreme occurs?

Page 3, line 24: It would avoid possible confusion to include a clarification at the end of Section 3 that despite having effects on the quantification of trends, these biases do not affect the conclusions in the study under scrutiny. When avoiding the discussed biases, there are still statistically significant increases in Rx1day and PRCPTOT in the dry (i.e. low-precipitation) regions.

Page 3, lines 26-30: To avoid the impression of bias, it is important to mention other definitions of 'dry' here that are also commonly used in the scientific literature.

Page 3, lines 31-33: Donat et al. provided a number of sensitivity tests, and also analysed Rx1day changes in the dry regions defined based on PRCPTOT (see their Supplementary Information SI4) – in this mask Scandinavia and the Netherlands are not part of the 'dry' class, but they still find increasing trends (and this is also the case after correcting for the biases discussed in Section 2). Please reword to avoid the impression of cherry-picking.

Page 3, lines 5 and 12: The statements about changes in spread of the spatial distribution do not seem to be relevant since only means are included in the analysis (not e.g. variance). These statements should be removed, or it should be explicitly stated that they are not relevant to the current analysis.

Page 4, lines 6-9: Over which time period where these alternative masks (2,3,4) defined? If not 1951-2010, you need to clarify that they may introduce the "regression

to the mean" bias.

Page 4, Line 9: What is the rationale behind including transitional regions when studying precipitation in dry regions?

Page 4, lines 15/16: large parts of these 'subsidence regions' with no or little precipitation changes are located over the ocean. Water availability can clearly not be a limiting factor here, so this is unrelated to the discussion of different definitions of 'dry'.

Page 4: Lines 17-21 give a hint of a balanced discussion, but unfortunately lead to a highly biased conclusion (lines 22-24), again appealing to what is supposedly 'commonly understood' and suggesting arid would be a conventional definition for dry.

**References**

Allan, R. P., Soden, B. J., John, V. O., Ingram, W. J., Good, P.: Current changes intropical precipitation. Environ. Res. Lett. 5, 025205, 2010.

Donat, M. G. , Alexander, L.V., Yang, H., Durre, I., Vose, R., Dunn, R., Willett, K., Aguilar, E., Brunet, M., Caesar, J., et al.: Updated analyses of temperature and precipitation extreme indices since the beginning of the twentieth century: the HadEX2 dataset, Journal of Geophysical Research: Atmospheres, 118, 2098–2118, 2013a.

Donat, M. G., Alexander, L. V., Yang, H., Durre, I., Vose, R., Caesar, J.: Global land-based datasets for monitoring climatic extremes, Bulletin of the American Meteorological Society, 94, 997-1006, 2013b.

Donat, M. G., Lowry, A. L., Alexander, L. V., O'Gorman, P. A., and Maher, N.: More extreme precipitation in the world's dry and wet regions, Nature Climate Change, 2016.

Greve, P., Orlowsky, B., Mueller, B., Sheffield, J., Reichstein, M., and Seneviratne, S. I.: Global assessment of trends in wetting and drying over land, Nature Geoscience, 7, 716–721, 2014.

Liu, C. and Allan, R. P.: Observed and simulated precipitation responses in wet and dry regions 18502100. Environ. Res. Lett. 8, 034002, 2013.

Sun, F., Roderick, M. L., Farquhar, G. D: Changes in the variability of global land precipitation. Geophys. Res. Lett. 39, L19402, 2012.

**Figure Caption** (complete caption as the online system seems to cut the caption after the second sentence):

**Figure 1:** Extreme precipitation changes in arid and humid regions. Time series of Rx1day (the annual-maximum daily precipitation) for dry/arid (a) and wet/humid (b) regions as identified by Greve et al., 2014. Area-weighted average time series are shown for HadEX2 and the ensemble mean and spread of CMIP5 simulations. Precipitation indices were first normalized by calculating annual values as a fraction of the 1951–2010 local mean before calculating the dry- and wet-region averages. Black lines, annual values from observations and ensemble mean; red lines, linear trend; blue dashed lines, 30-yr averages for 1951–1980 and 1981–2010; grey shading, ± one ensemble standard deviation. dRx1day indicates the difference between the averages during 1981–2010 and 1951–1980; slope is the linear trend Sen-slope estimate (unit, decade−1); and the p-value is the trend significance using a Mann–Kendall test. (c)

The mask indicates the locations of the grid cells contributing to the average of the dry (red) and wet (blue) regions, and the number n of grid cells contributing to the area averages of dry and wet regions is given. Land grid cells that are less complete than 90

―――――――――――――――――

[Figure]

[Figure]

**Fig. 1.** Extreme precipitation changes in arid and humid regions. Time series of Rx1day (the annual-maximum daily precipitation) for dry/arid (a) and wet/humid (b) regions as identified by Greve et al., 2014.

---

## Referee Comment (RC4) · Anonymous Referee #4 · 30 Oct 2016

Overall, I am pleased with the topic of the Sippel et al. paper, which is an evaluation and criticism of some of the methods used in the Donat et al. 2016 paper More extreme precipitation in the world's dry and wet regions. This is the type of check-and-balance that keeps our science robust. Sippel et al. address two main criticisms of the Donat 2016 paper, (1) the introduction of a statistical bias when the rainfall data is normalised, and (2) the introduction of another statistical bias based on the regions that are selected as "dry" and "wet" regions. The overall flow and readability of the paper was dense, but not unfollowable. However, I understood the context of the paper, and the authors' intention, much better after I read the Donat et al. 2016 paper. The authors could use more precise wording to clarify that the methods used were done to recreate the

results from Donat et al. 2016.

On the topic of the introduced bias from normalising the data; the process of normalising data is pretty common and ensures that areal averages are not dominated by very wet regions. However, this needs to be done with care. The authors unpack and clearly describe the statistical changes that are introduced from the normalisation process. I liked the illustrative example found on page three in lines six through 11 and the quantification of the bias (%) in appendices A and B provided good support for the argument. (Although it isn't clear why these are included as appendices and not tables in the paper). Furthermore, the authors do well to point out the changes that arise by using different reference periods to deconstruct the data (i.e. Figure 2). I note that it was not really clear from reading the Donat (2016) paper why they used the 1951–1980 period to normalise the data.

I don't completely agree with the argument for selecting dry regions. The criteria and thresholds used to define a dry region are very subjective. As Sippel et al. point out, precipitation alone is not enough to determine if a region is wet or dry–e.g. at very high latitudes where even small amounts of rainfall can exceed the potential evapotranspiration. However, the criteria used are dependent on the question to be answered. If the question to be answered is, "How are global precipitation patterns changing?" then an analysis of precipitation alone would be sufficient. If you are trying to address, "Are wet/dry regions getting wetter/drier?" then the hydrology/aridity or climate classification of the region would need to be considered.

The authors quantify the "regression to the mean" bias (as shown in appendices A and B) that arise by defining dry areas as the lowest 30%. The authors further demonstrate that by using the Köppen classification and the Greve (2014) definition that the large trends found by Donat et al. are dramatically minimised. I think this argument is a moot point because, as other reviewers have already pointed out, the HadEX2 dataset does not have data over the world's driest regions (e.g. the Sahara, Western Australia) or some of the wettest regions (e.g. the Amazon or the Maritime Continent region).

A global analysis or precipitation extremes or precipitation trends using HadEX2 data would deliver incomplete results.

Specific comments: 1. Page 4, line 12: mentions a two-sided trend test. Is this the same as the Mann–Kendall test used by Donat at al. and mentioned in the caption of figure 3? It is not really clear in the body of the text why or how this test was chosen.

2. Appendix A, Figure 0, caption: check the spelling of Köppen. This figure was hard to understand. After reading the caption a few times I understood that it is basically built as a table with the first (left) column being the PRCPTOT data and the second (right) column being the Rx1D data. It would be nice to have the rows/columns clearly labelled.

3. Figure 2: The caption mentions red lines. The lines look orange to me.

4. Figure 3: I found this figure very difficult to understand. There is a lot of information that is overlayed on other information. The grey text is too light against the white background.

5. Your methods for producing this graph (grey and black lines) are not clear. You mention the grey lines have been corrected for "statistical artefacts"; I could not find this correction explained anywhere. Which artefacts have you corrected for? Is it the bias from the normalisation? Likewise, the process for producing the black line, or removing the incomplete data, is not explained.

6. The label on the first row of graphs mentioned the Köppen–Geiger climate classification, but the caption references Köppen (1900). The Köppen–Geiger classifications were not published until Geiger (1954 and 1961). Kottek et al. 2006, which was mentioned in the text, is of the Köppen–Geiger classifications. Should the caption reference Kottek et al. 2006 rather than Köppen (1900)?

7. Are graphs 3.e and 3.f from the Greve data, dry+transitional regions? It is not clear from the caption.

---

## Author Comment (AC1) · 21 Nov 2016

**Response to RC1:**

**S. Sippel et al., 2016**

This manuscript highlights two concerns with the recent analysis presented in Donat et al (2016), who reported increasing trends in extreme precipitation and total precipitation in dry regions around the world. The two concerns raised by the authors with the analysis of Donat et al (2016) are valid. Here the authors present a re-analysis of the Donat et al (2016) work using more appropriate methodologies and find the results of Donat et al (2016) are highly dependent on the methodology they adopted, which has significant implications for the conclusions of Donat et al (2016). Overall, this manuscript presents an important critique of Donat et al (2016), which is highly relevant to the general scientific community. I believe the manuscript will be worthy of publication following moderate revision to improve the clarity of the manuscript, as it is hard to follow at times.

**We thank the reviewer for the positive evaluation of our manuscript. We have taken the reviewer's suggestion about improving the clarity of the manuscript into account as specified below in detail.**

My key comments / suggestions are as follows.

The overall style of this manuscript is abbreviated and densely packed. In fact, some sections are difficult to follow as helpful explanations are not provided. Figures and Tables are cited, but little accompanying explanation is provided to help the reader understand and interpret them. The current manuscript style is like a very compact 'communication arising'. I think this style provides the key messages, but it is very difficult to follow the technical argument in places. Also, why are Appendix A and B not just normal Figures and Tables, like the other Figures and Tables? Why the separate Appendices? I recommend you move this material from the Appendices into the paper.

The key points made in this manuscript are fine, but the explanation accompanying the Figures and Tables requires expansion to improve the readability of the manuscript. At times I found it difficult to know how each Figure and Table slotted into the overall story; not because the material is irrelevant, but because a context for the material was not provided. There is a lot of interesting material in the Figures and Tables, which is hardly covered in the text. Expanding the explanations around the Figures and Tables will guide the reader through this important material and significantly increase the accessibility of this research.

We understand the reviewer's concerns about the very dense style of writing and presentation. Indeed, the reviewer is correct that our objective was to provide the key messages in a very dense format. To account for the issue rightly pointed out by the Reviewer and to improve readability and clarity, we extend the explanations in the revised manuscript, and improve the embedding and context of figures and tables. We will also move the tables into the main manuscript and provide the respective context.

For example, we extend the explanation of the normalization-induced bias (p. 3, l.1-14) by the following paragraph that provides more context to Fig. 1:

*"This issue is illustrated in Fig. 1 for an artificial dataset that consists of n = 104 time series (e.g., 'grid cells') that are drawn randomly and independently from a Generalized Extreme Value distribution (GEV, Coles et al., 2001). The GEV distribution provides an asymptotical limit model for maxima derived from a sequence of random variables with fixed block size (Coles et al., 2001, e.g. the annual-maximum daily precipitation,), and is therefore appropriate to illustrate this issue. Normalising each time series in the artificial dataset with its mean in the first period yields a spatial 'reference period distribution' that is different from the spatial 'out-of-base period distribution' (and from the original GEV distribution, Fig. 1a), e.g. resulting in increased spatial averages in the out-of-base period (Fig. 1b). Furthermore, the normalisation procedure induces a considerable increase in the variance, skewness and higher statistical moments in the spatial distribution in the second period (see e.g. Fig. 1a), which would be of relevance if higher statistical moments (changes in spatial variance, etc.) would be studied. The reason for this observed difference lies in the fact that the sample mean (derived from the reference period) is a dependent estimator for the reference period time series, but a (virtually) independent estimator for the time period that lies outside of the reference period (Zhang et al., 2005; Sippel et al., 2015)."*

Please note that in addition we include an analytical argument to derive an approximation for the expectation value of the normalization-induced bias in the revised version of the manuscript (please see attached pdf-file). However, in order to not compromise readability, we will include this in an Appendix or as Supplement. This argument thus provides an explanation as to why the biases are systematically positive outside of the reference period, and shows that this bias is proportional to the ratio between mean and standard deviation. We believe this is useful, because this type of reference period normalization is indeed common in many studies (not only in Donat et al.), and therefore the analytical approximation might provide a useful -first order- estimate on the magnitude of the expected bias.

Furthermore, to improve readability, we include an additional simple example that illustrates the second statistical issue, i.e. the "regression to the mean" effect, because the explanation as it stands now (p.3, l. 14-l.19) might not be immediately clear to all readers. Hence, we will add a simple example after l. 17:

*"To illustrate this issue, recall the conceptual two-region example quoted above, where variation between the two available time periods would be entirely due to random causes. If any of the two periods would be chosen to stratify the dataset in one dry and one wet region, this would result in opposing changes (i.e. dry gets wetter, wet gets drier) in the second period."*

Minor comments

Page 3, line 18: "the dataset will result". Are you 100% certain it will result in a higher probability for wetter (drier) conditions. Or is a better word to use here "may" result. I think this paragraph would benefit from an expanded explanation of the statistical bias being discussed as it is not easy to follow.

**Thanks for the comment, we agree the paragraph needs to be expanded to be made more clear. In a statistical sense (i.e. assume all grid cells to be random variables and assume many of them) we are certain that "selecting from the dry (wet) end of the spatial distribution in one subset of the dataset will result in a higher probability for wetter (drier) conditions in the remaining years". This is because random variation in the reference period plays a role and will lead to the regression to the mean phenomenon. But of course this only holds if there is not a systematic (global) shift outside the reference period. We will rephrase the sentence in a revised manuscript to make clear that it is a statistical expectation, rather than 100% certain.**

Appendix B second Table: Replace "Rx1day" with "PRCPTOT" in the wet and dry regions.

**Thanks for reading thoroughly. Correct, and has been changed.**

Figure 1: you need to improve the explanation of this Figure. The illustration provided in the text (page 3, lines 6-13) was excellent, but the connection to Figure 1 was not obvious.

**Again, thanks for pointing this out. As noted above, we have included an extended explanation for the figure (which will be inserted in p.3, l. 13)**

Figure 3c, 3d: are the p-values for the one-sided and two-sided trend tests reported correct or have they been switched?

**Again, thanks for reading thoroughly. The p-values have been switched indeed and we have corrected it.**

Tables 1 & 2: Explain what Period Inc. means.

**"Period Increment" means the period changes between the first**

(1951–1980) and the second (1981–2010) period. We will explain this better in the revised manuscript.

---

## Author Comment (AC2) · 21 Nov 2016

**Response to RC2:**

**S. Sippel et al., 2016**

Review of Sippel et al., 'Have precipitation extremes and annual totals been increasing in the world's dry regions over the last 60 years?'

This paper (which can effectively be considered as a comment on the Donat et al (2016) paper) raises two issues with the Donat et al (2016) (hereafter D2016) paper — the way in which spatial averaging has been used and the way in which dry regions have been defined.

Both of these are legitimate concerns. However, in my view both this paper and D2016 miss what I think is the main point with respect to the definition of dry regions — namely, that most of the world's driest regions (in particular, almost all of the Sahara and the Middle East) are excluded because of a lack of data. (Similarly, many of the world's wetter regions in South America, equatorial Africa and southeast Asia are also excluded). Any definition, whether it is the one used in D2016 or in this paper, is likely to give an unrepresentative sample of the world's dry regions given that the data availability is largely confined to North America, Eurasia and parts of Australia. Put another way, the HadEX2 data set in its current form is not capable of providing a fully representative sample of the world's dry regions, which is particularly important given that there is no reason why we would expect tropical arid and semi-arid zones (e.g. the Sahel), subtropical deserts (e.g. southwest US) and high-latitude low-precipitation regions to have similar long-term trends. A casual reader encountering either this paper or D2016 would expect the papers to be covering a very different range of areas to that which they actually do.

We thank the reviewer for highlighting this issue. The reviewer is of course correct (and the other reviewers who have pointed this out), data scarcity is a large caveat in analyses about precipitation characteristics in "the world's dry regions".

However, we believe that scarcity of data alone should not prohibit scientific analyses on precipitation characteristics such as the one by Donat et al. (or ours) in those dry regions where data are available. Therefore, in the revised manuscript we will make this point more clear in the text, and we particularly highlight that "dry regions of the world" actually implies "dry regions of the world where data are available". Moreover, to make this point clear to casual readers also, we will include Appendix A (Figure 0, the maps of the dry region definitions) in the main manuscript (as Reviewer #1 also recommended).

(I would view both this paper's method and the D2016 method as being reasonable ways of defining dry regions — the issue is that neither is representative given the gaps in the data set).

**Again, we agree with the Reviewer: Our manuscript was not intended to reject the definition used by Donat et al (please see also our reply to the short comment by M. Donat). We simply intended to illustrate that for the overall change in dry region extreme precipitation, it does make indeed a relatively large difference (i.e. there is a large sensitivity to how one defines a dry region) if one analyses extreme precipitation (Rx1d) trends in "regions of light extreme rainfall" (that could be humid throughout the year, cf. our response to the short comment by M. Donat), or whether one studies trends in "dry" (arid) regions.**

Averaging precipitation indices is another challenge — whilst the averaging period (as mentioned in this paper) is one issue, another is whether it is appropriate to average values from a distribution which is bounded below by zero and highly non-Gaussian. If one averages absolute values, area averages are likely to be dominated by the wetter areas; if one averages normalised values, there will be much more volatility in the driest areas. (Somewhat ironically, the fact that the HadEX2 data set excludes most of the world's really dry areas averted a bigger problem here — in climates where mean annual values are, say, below 10 mm, annual totals in excess of 1000% are plausible, which would completely overwhelm less variable climates in a spatial average).

**The Reviewer raises an interesting and highly relevant point – and we have investigated in detail as to whether the methodological issues due to reference period normalization and subsequent spatial averaging would have been worse in arid regions where there is currently no data available.**

**First, we would like to point out that we agree with the Reviewer: Averaging absolute values would lead to wetter areas dominating the response, and that is why a normalization of some sort is required. Second, random variation in dry regions and the normalization procedure with subsequent spatial averaging (as illustrated by the Reviewer in the 1000% example) is precisely the root cause of the biases identified in our paper. These biases are related to the length of the reference period, but also to the ratio of mean:standard deviation (or location:scale in a GEV distribution) in the Rx1d time series (as implied by the Reviewer; and we will make this point clearer in the revised manuscript). Hence, it can be seen analytically that 1) these biases are systematically positive outside the reference period, and 2) the biases scale with the change (i.e., trend) in a multiplicative way (please see the attached pdf-file, we intend to include this material in the Appendix of a revised manuscript); and it is also**

possible to derive a (first-order) analytical approximation of the expectation value of the biases as a function of the ratio mean:sd, and the length of the reference period. The analytical approximation allows to derive some estimates of the magnitude of the biases if a grid-cell scale normalization is followed by spatial averaging. We believe that this is useful to have because the applied methodology is quite common and not specific to the Donat et al. study - for instance similar data processing methodologies based on fixed reference periods are used to bias-correct relative precipitation anomalies from climate models to some observational datasets (Hempel et al. 2013, ESD, doi:10.5194/esd-4-219-2013), or observational datasets are derived based on station anomalies from a fixed baseline (Harris et al., 2014, doi/10.1002/joc.3711). However, we also note that in many real-world cases, the introduced biases will be small because the ratio of mean to standard deviation be high (e.g. in humid regions), but nonetheless it is important to note that this type of bias exists.

However, having the analytical approximation at hand we can assess how the expectation values of the biases would differ between regions, in particular whether the normalization-induced bias would become worse if more stations from data-scarce arid regions would be available. To do so, we downloaded all available arid-region station data from the GHCNDEX database (http://www.climdex.org/sewocs.html), and disregarded all stations with less than 30 years of data. Subsequently, we stratify these stations according to whether they lie in regions with or without data availability in the HadEX2 dataset, and calculate the long-term mean and standard deviation of each individual station.

The sample mean of the stations are indeed lower in data-scarce regions (Fig. 1a), i.e. arid regions without data in HadEX2 (and those stations in arid regions without data that lie in Africa only, Fig. 1a) tend to systematically receive less extreme rainfall (Rx1d). However, we notice that the ratio of mean:sd parameters is approximately similar in these regions (which would indicate a similar magnitude of the biases). An approximation of the expected bias in data-scarce regions (Fig. 3d) indicates that the expected bias would be slightly higher (+0.67\% vs. +0.71\%, +0.85% in stations that lie in Africa).
Hence, we conclude that the reviewer is correct, the systematic biases would be slightly larger in data-scarce arid regions if data would be available there, but it would not be a completely different story. We do not intend to include this analysis in the revised manuscript (because it might lead a bit too far away from the actual story thereby compromising readability and clarity), but we will include a comment in the discussion of the normalization-induced biases (p.3, l. 1-14) to the fact that these biases will be higher if the mean:sd ratio is lower (as seen from the analytical approximation),

which one might expect in very dry, currently data-scarce regions.

[Figure]

Figure 1: **(a) Sample mean, (b) ratio of mean:sd at individual stations of the GHCNDEX (stations-based) dataset in arid regions ("Arid w. data": Stations in arid regions with data in HadEX2; "Arid no-data, all": Stations in arid regions without data in HadEX2; "Arid no-data, Africa": Stations in arid region in Africa without data in HadEX2). c) Expectation for artificial increase according to Eq. A8 (with only the first term of the Taylor series) in attached pdf-file. d) Map of arid regions of the world (Greve et al. 2014): Orange regions indicate data availability in the gridded HadEX2 and GHCNDEX datasets, whereas grey areas indicate data gaps. Individual stations in grey areas are denoted by black dots.**

In my view, it would be better not to try to do spatial averages at all, and instead report using indicators such as the % of gridpoints which show significant positive/negative trends. That said, if you are going to average precipitation indices, then this paper has identified a genuine issue with the D2016 methodology.

**We agree with the reviewer that in general reporting station-based indicators would be at least equally important. However, in order to compare quantities to climate models, or to study globally aggregated quantities relative to a common baseline (for which there seems to be a demand, see e.g. Hansen et al. 2012, Huntingford et al. 2013, Seneviratne et al. 2014, or similar papers that outline globally aggregated temperature characteristics), we believe that this approach can be indeed useful, too.**

In summary — I think this paper accurately documents valid issues with the D2016 paper, and as such I think it is appropriate for publication, but I also think it would be improved if it engages substantially more with the issues identified above.

**We thank the Reviewer for the positive evaluation and hope that we have addressed your concerns properly.**

---

## Author Comment (AC3) · 21 Nov 2016

**Response to RC3:**

**S. Sippel et al., 2016**

The authors examine the robustness to choices made in the analysis of a recent analysis of observed trends in precipitation in dry regions around the world. In general I quite like this, as results of studies are usually interpreted beyond the specific experiment design of the analysis, and so this paper performs the important task of determining the extent to which it is possible in the case of observed precipitation trends in dry regions. However, I think there are a couple of additional aspects to this that the authors have not considered, as well as one important syntactic issue, that I believe need to be addressed before publication.

**Thanks for the positive evaluation!**

First, the motivation you frequently mention is for informing adaption decisions. For that motivation, though, it is not clear to me that what you do in terms of normalising to the full period is necessarily any more appropriate than what Donat et alii (2016) did. Many decisions are based on climatic or hydrologic data from a specific time period, for instance in the case of international treaties allocating water on an international river. Thus adaptation decisions need to be made with respect to divergence from that reference baseline (ignoring non-climate stuff). So while e.g. you may be correct that there has been no actual trend in precipitation totals, say, that does not necessarily mean there has not been a trend as measured by stipulated monitoring procedures used in many decision-making settings. Cast another way, we have the same problem in dealing with future climate change: projections are based on, say, the full historical period you use, but that does not include the future itself. I expect you are not arguing that we cannot make useful projections of future climate change simply because we have not monitored the future yet. In this context, I laud your effort because you high-light the sensitivity to this point, but I think it is important — and entirely consistent with you consideration of robustness — to emphasise that there is not necessarily a single "correct" answer.

**Thanks for this point – indeed we are not arguing that no useful policy choices can be made just because the future is not part of the reference period. This would be somewhat nonsensical and clearly overstate the problem. However, for the latter problem, please note that it is not just an issue of random variation of the reference period – it is indeed a systematic bias that is 1) positive outside the reference period, and 2) scales with the relative change in the time series. The expectation value of the bias is a function of the reference period length, and the mean:standard deviation ratio (or location:scale ratio and shape parameter in a GEV). Please**

**have a look at the analytical approximation (attached pdf-file), which we intend to include in the Appendix or Supplement of a revised manuscript.**

**In many practical settings the bias might be small (e.g. if the ref. period is long enough, or the mean:standard deviation ratio is high), but in other cases it might be relevant for example because the trends in the quantity of interest might be small. In our particular case, we have chosen to normalize to the full period because this avoids the bias. However, in a hypothetical case where one would be bound to a given baseline period (for instance in your example where an international treaty would define climate change or its impacts relative to a specific baseline), one could still take our analytical results and estimate the expected magnitude of the biases relative to the observed trend. Thanks again for this hint, and we will emphasize in the revised manuscript that there is no correct answer for the choice of the reference period, but with either avoiding a "reference - non-reference breakpoint" or by analytically adjusting for it using e.g. our approximation (which also works if there are in-stationarities in the time series outside of the reference period).**

Second, in terms of all of the discussion about what constitutes a "dry" region, the most striking thing to me is that none of the definitions of dry regions you consider include what I think of as prototypical dry regions: the Sahara, the Saudi Peninsula, Central Asia (particulaly for Rx1D), southwestern Africa (other than South Africa), western Australia, northern Mexico (for Rx1D), nor the driest areas of South America (for Rx1D). The reason for this of course is monitoring coverage, but given the absence of all of these regions (the Sahara!) I do not think these results can plausibly be considered as being indicative of how precipitation is actually changing over the world's dry regions. Again, I consider this a point about robustness that is entirely consistent with your paper, but it most certainly needs to be acknowledged/noted/highlighted.

**Yes, we agree (please see also additional analyses in the arid data-scarce regions and more detailed reply to Reviewer #2 who made a similar comment). In a revised manuscript, we will this point in the text, Abstract, and Conclusions, and also move Appendix A (Figure 0) in the main manuscript to ensure that even a casual reader will encouter the aridity and dryness maps, including the gaps in spatial coverage.**

Third, on the syntactic side, while the title refers to precipitation and it appears to be precipitation you are actually analysing, within the text this is generally referred to as "rainfall". Please clarify which you are examining, as these are certainly not identical for annual totals (and, if

defined in certain ways, for heavy extremes) in many of the regions you examine.

**Thanks for this point. "Rainfall" and "rainfall extremes" are indeed used erroneously and we will change it to precipitation in a revised manuscript.**

Specific comments:

page 1, line 1 The title says you are examining precipitation extremes and annual totals, but here you indicate it is rainfall. Which is it? It seems to be a precipitation dataset you are using, so it looks like the usage of "rainfall" is wrong?

**Yes it is, please see above.**

page 2, lines 24-25 If they are being underestimated, then it sounds like the errors are not completely cancelling, right?

**Not completely, correct. Will be rephrased in a revised manuscript for clarification.**

page 2, line 25 "These results": Which, Donat's or yours?

**Actually both, but we'll clarify.**

page 2, lines 25-26 I think such an assertive statement concerning the decision-making processes utilised in dry regions requires some supporting evidence, e.g. to other research on decision-making in those regions.

**That is true. We have rephrased our statement to reflect that an accurate quantification of change in precipitation characteristics (which includes monitoring, etc.) is important because it is simply a prerequisite to be able to make climate change adaptation decisions.**

page 3, line 9 "in both time periods" -> "over the combined periods"

**thanks.**

page 4, lines 21-22 This is a case where if you are considering rainfall, and not precipitation, then indeed North-East Siberia is rather dry.

**This is true, but we have changed the discussion to discuss precipitation; and discuss "dryness" in terms of "low precipitation" (precipitation alone), "low annual-maximum rainfall", and "dry in terms of water availability", i.e. supply and demand (arid climate).**

page 4, lines 25-26 I do not believe that Fischer and Knutti (2015) studied the decision- making processes used by those

involved in responding to climate change, and in particular
what they considered "relevant" information for informing
those processes.

**Yes, true. We have rephrased to better reflect what we mean
and removed the reference to the paper mentioned.**

page 6 "Figure 0" should have a different identifier.

**Yes, and the figure will be moved into main manuscript.**

page 6, caption Can you confirm that for only different
between columns for the lowest two rows is the mask? I found
this caption confusing, for instance with the distinction
between the columns being introduced only halfway through.
Subtitles on each panel could help.

**The difference between the columns for the lowest two rows is
the mask and the data availability of HadEX2 for PRCPTOT
(left) and Rx1d (right). We'll clarify the caption.**

Figure 2, caption line 3 By "red lines" do you mean yellow?
Tables 1 and 2 What does "Period Inc. (%)" mean?

**Thanks for reading thoroughly! Red should be orange, and
"Period Inc." means the relative period changes between the
first and second period (i.e. 1951–1980 vs. 1981–2010). Will
be corrected/clarified in a revised manuscript.**

Tables 1 and 2 Why do the "Sample size" values differ? Aren't
all the trends calculated over the same number of years?

**By "sample size" we mean the number of grid cells over which
the spatial averages are taken. We'll clarify.**

Table 2 There is one trend values listed as "<0.000". Why do
you not give the numerical value for a negative trend? This
one is interesting, because it is the only significant
negative trend.

**Oh, sorry. The trend is actually almost exactly 0 and not
significant. Will denote this one as "0.000".**

---

## Author Comment (AC4) · 21 Nov 2016

**Response to SC1:**

**S. Sippel et al., 2016**

The authors scrutinize a recent study (Donat et al. 2016) that reported increasing trends in precipitation extremes and annual totals in the world's dry regions, as defined by precipitation amounts. The authors (1) suggest that the results of the scrutinised study were biased owing to choices of the reference period, and (2) discuss that the findings depend on how 'dry' regions are defined.

We thank the authors for pointing out the statistical issue related to the reference period which is now addressed in a Corrigendum (submitted to Nature Climate Change on 12th September 2016). Importantly, this statistical issue does not affect the major conclusions of the scrutinised study, a point that should be made clearly in the current manuscript. However, the remainder of this manuscript, in particular the discussion related to the definition of dry regions, is biased, inconsistent, ambiguous (misleading), and incomplete as outlined below. Therefore the manuscript needs to be carefully revised before publication.

**We thank M. Donat for the partly critical and partly positive comments on our manuscript. We appreciate that pointing out the statistical issues and sensitivities related to the data-analytical tools applied in the original study are welcomed. Please note that partly in response to reviewer comments (who stressed that the normalization-methodology is indeed relatively common), we include some analytical approximation (see pdf-file) that allows to derive analytical estimates of the biases as a function of reference period length and the parameters of the distribution. We hope these estimates are considered as useful, and we intend to include this material in a revised manuscript as Supplement or Appendix.**

**We also acknowledge the critical comments regarding the definition of dryness, and have carefully addressed the comments raised.**

*Biased:* The current manuscript claims that the only valid definitions of wet and dry regions are those based on surface water availability, referring to what is 'commonly understood' or 'conventional'. However, in everyday language it is common to use 'wet' or 'dry' to refer to high or low precipitation for both regions and times of year. Furthermore, in the scientific literature there are numerous related studies that have defined wet and dry solely based on meteorological parameters such as precipitation (e.g. Allan et al.,2010; Sun et al., 2012; Liu and Allan, 2013), and these are ignored in the current discussion and should be included in a revised manuscript. The current manuscript, therefore, appears biased in that it is largely based on a claim that only a particular

definition of dryness is valid, when several other definitions are in common use.

The purpose of the section on the definition of dryness was not to claim superiority of any particular dryness definition. In contrast, the main idea behind this exercise was to test how sensitive trend slopes or period changes are to alternative definitions of dryness, given that early climatological research had used the word "dry" in terms of water availability rather than precipitation totals alone (see our manuscript). However, we do not claim that this is superior. We have made this point clearer in the manuscript, and we also acknowledge that the three papers cited above use a dryness definition based on precipitation totals (annual or monthly climatology), similarly as Donat et al (2016) do in their original study for annual precipitation totals (PRCPTOT).

However, for annual-maximum daily precipitation (Rx1d), we believe it is crucially important to consider this additional point: In contrast to the studies cited above, in the original NCLIM study by Donat et al, the definition of "dryness" has been made based on the annual-maximum daily precipitation amount (i.e., Rx1d). This means that any region with a relatively modest 1-day extreme rainfall would be considered as "dry". This is in contrast to the three papers cited above, because Rx1d is not necessarily strongly related to precipitation totals. For example, the potential for very strong convective rainfall in high Northern latitudes (e.g. Scandinavia, Siberia) might be limited, therefore resulting in moderate annual-maximum daily precipitation, while the region could still be "wet" throughout the year (either in terms of precipitation totals, or in terms of water availability, or both). To illustrate this example, please see the plot below computed from the original HadEX2 data (1951-2010 means, using the 90% threshold for NA-removal):

While there is clearly a relationship between PRCPTOT and Rx1d, we find that only 22% of the "dry" grid cells according to the maximum-annual precipitation definition are also "dry" given annual precipitation totals (see plot below). Hence, while we do understand the notion of exploring (spatial) extremes in the "HadEX2 data space", it becomes an issue of semantics here: We argue that regions with low annual-maximum precipitation should simply be called for example "regions with low maximum precipitation" rather than "dry" as this might lead to confusion (e.g. if cited in IPCC reports, reported by the media, etc.). Similarly, if compared with aridity, the difference between a definition based on precipitation totals, rainfall extremes and aridity becomes very clear (see figures below).

In summary, we have changed the manuscript such that it becomes clear that we simply explore alternative definitions

**of dryness, we do not claim superiority or call it "common understanding", etc., and we now also state that definitions based on precipitation totals had been used previously in the literature.**

[Figure]

**Figure 1: Relationship between PRCPTOT and Rx1d in HadEX2**

[Figure]

**Figure 2: Relationship between a) PRCPTOT and Aridity, and b) Rx1d and Aridity in the HadEX2-GHCNDEX merged dataset. Potential evapotranspiration is taken from the CRU-TS3.23 dataset (Harris et al., 2014).**

An important point that emerges from this discussion is that it is desirable to specify more clearly which type of definition of dry and wet is being used in studies of climate change. Indeed this is something the current manuscript could do better; see 'ambiguous' section below. We suggest to the authors that they make the conclusion of their paper and abstract a call for more specificity in the use of 'dry' and 'wet' in climate-change studies. For example, one could refer to 'meteorological' or 'hydrological' wet and dry regions, by analogy with the standard definitions of 'meteorological' or 'hydrological' drought. This would be of greater value than arguing that only one type of definition is valid.

**Thanks for this important point, we agree. We have shaped our manuscript more in this direction and distinguish between precipitation totals, regions of moderate annual-maximum precipitation (for Rx1d) and aridity-based definitions.**

*Inconsistent:* The analysis in Section 3 is likely affected by the same "regression to the mean" bias discussed in Section 2, because the dry-regions masks that include water demand were not defined over the entire study period 1951-2010.

**We do not think this is the case: In Section 3, we use two different dry-regions masks, one based on Köppen-Geiger, and another one based on Greve et al (2014). The former had been derived in 1900 without any data from 1951-2010, so it cannot be affected. The latter dry region mask has been derived from the 1948-1968 period, but based on a very large number of dataset combinations (77) at the gridbox level and based on the aridity index (i.e., for two different variables: Potential evapotranspiration, Ep and precipitation, P). Therefore, estimates of the annual-maximum precipitation (Rx1d) should be virtually independent, as this variable is not (at all) related to Ep, and only weakly related to P. Similarly, the "regression to the mean" problem should be virtually eliminated also for precipitation totals due to the different combinations of datasets that inhibit random variation within one dataset, and because the Greve et al dry-region mask "is in good agreement with the commonly used standard climate-classifications of Köppen-Geiger" (Greve et al, 2014, NGEO). Any remaining "regression to the mean" issue in PRCPTOT would lead to a positive bias in the trend slopes and period changes relative to the Köppen-Geiger mask, but in fact the trend slopes and period increments obtained with the Greve-mask are smaller than those obtained with the Köppen-Geiger mask (see Table 2 in the manuscript). Hence, we conclude that "regression to the mean" is not an issue in our manuscript.**

*Ambiguous:* The current text uses 'dry' for different concepts, and this is likely to confuse readers. To avoid confusion, the authors should specify whether they are talking about 'low-precipitation' or 'arid'/'water-limited' regions. This is particularly problematic e.g. in the Abstract lines 3-5 where dry is defined in terms of water availability but then immediately used to refer to the scrutinised study in which dry means low precipitation. Similarly in the introduction it needs to be specified which concepts of 'dry' the authors refer to in each case.

**Thanks for this point. As indicated above, we believe this is a very good idea and we have incorporated it in the manuscript.**

*Incomplete:* The main reason why Sippel et al. don't find a (statistically significant) increase in Rx1day in arid regions seems to be related to scarcity of data. It is unfortunate reality that arid regions are insufficiently covered by observations. Aggregating only over a few grid cells results in relatively noisy time series, so that — despite a positive trend slope — the p-value of the applied trend test is too high to reject the null hypothesis of 'no change'. A relatively easy attempt to optimise spatial coverage by merging the two existing datasets HadEX2 (Donat et al., 2013a) and GHCNDEX (Donat et al., 2013b) gives a few additional grid cells with data in arid regions. Aggregating over this just slightly improved coverage results in a more robust trend estimate in observations and in the CMIP5 ensemble mean (Figure 1). This suggests that a major uncertainty when analysing precipitation changes in arid regions comes from the limited availability of observations. Also, if using the complete coverage as provided e.g. by the ensemble of CMIP5 models as used in Donat et al. (2016), the authors would find statistically significant increases in ensemble mean over the arid regions (not shown). Therefore we assume that the main reason why Sippel et al. conclude there is 'no significant increase in heavy precipitation' in arid regions is related to the scarcity of observations.

**Thanks. We agree that the scarcity of observational coverage and resulting noisy time series can be a major obstacle to detect significant trends. As suggested, we have merged the HadEX2 dataset with the additional GHCNDEX dataset that contains data. This results in a minor upwards change in trend slopes and period increments, and that several (but not all) trend slopes are indeed significantly increasing. Hence, we report these additional results in the revised manuscript. For example, our revised Conlusion reads:**

*"Monitoring and an accurate quantification of trends in meteorological risks in a rapidly changing Earth system is a prerequisite to informed decision-making in the context of climate change adaptation (IPCC 2014). Therefore, short reference periods that are defined on a subset of the available dataset for normalisation or data pre-processing purposes should be avoided, as this procedure inevitably introduces biases (Zhang et al., 2005; Sippel et al., 2015). In the present study under scrutiny, these statistical effects reduce the reported trend slopes and period changes by up to 40%, but the direction of the overall signal remains unchanged (i.e. increasing trends in Rx1d and PRCPTOT in regions of moderate extreme precipitation and low annual totals, respectively).*

*Furthermore, the definition of a `dry region' induces considerable uncertainty in quantifying changes in precipitation extremes or totals. If dryness is defined based on water supply and demand (i.e. aridity), we find a systematic and significant reduction of trend slopes and period increments in annual-maximum extreme precipitation and precipitation totals, which yields some significant and some in-significant (depending on precipitation characteristic, pre-processing, and specific dryness definition considered) but exclusively positive trend slopes (Table~\ref{table2} and Table~\ref{table3}). Hence, overall we confirm an indication towards increases in both metrics in the world's dry regions.*

*However, as a caveat to the present study, it is important to stress that a large part of the world's dry regions, such as large arid and semi-arid regions in Africa, the Arabian peninsula, and partly South America are not covered by monitoring datasets that are available at present. This fact highlights the importance of consistent, long-term monitoring efforts, data quality control, development and maintenance of long-term datasets (Alexander et al., 2006; Donat et al., 2013a,b), and also emphasises that the results reported here should be regarded as indicative only for those arid regions where there is data available at present.*

*In summary, understanding and disentangling on-going changes in precipitation characteristics in the world's dry regions remains a research priority of high relevance. In this context, our paper demonstrates that 1) data pre-processing methods can introduce substantial bias, and 2) trends and period changes can be sensitive to the specific choice of dryness definition that is used; therefore we urge authors to be considerate and specific regarding both choices and to consider associated uncertainties. "*

**Specific comments:**

Page 2, line 3: 'if there is enough moisture available' — do you mean annual average moisture availability? Or seasonal? Or on the day the rainfall extreme occurs?

**By "if there is enough moisture available", we mean enough moisture available for the extreme precipitation to occur, i.e. sensu e.g. Trenberth (2003). However, this sub-sentence is not necessary for the meaning and we have clarified the first sentence.**

Page 3, line 24: It would avoid possible confusion to include a clarification at the end of Section 3 that despite having effects on the quantification of trends, these biases do not affect the conclusions in the study under scrutiny. When avoiding the discussed biases, there are still statistically significant increases in Rx1day and PRCPTOT in the dry (i.e. low-precipitation) regions.

**This is correct and had not been disputed in the original manuscript. However, to make this point crystal-clear, we have added a clarifying sentence as suggested.**

Page 3, lines 26-30: To avoid the impression of bias, it is important to mention other definitions of 'dry' here that are also commonly used in the scientific literature.

Page 3, lines 31-33: Donat et al. provided a number of sensitivity tests, and also analysed Rx1day changes in the dry regions defined based on PRCPTOT (see their Supplementary Information SI4) — in this mask Scandinavia and the Netherlands are not part of the 'dry' class, but they still find increasing trends (and this is also the case after correcting for the biases discussed in Section 2). Please reword to avoid the impression of cherry-picking.

**As pointed out above, we have extended the discussion of the definition of dry regions: This discussion mentions now that also dryness definitions based on precipitation totals are in use, and discusses Scandinavia and the Netherlands only in terms of the dryness definition (thus, there should not be the impression of cherry-picking).**

Page 3, lines 5 and 12: The statements about changes in spread of the spatial distribution do not seem to be relevant since only means are included in the analysis (not e.g. variance). These statements should be removed, or it should be explicitly stated that they are not relevant to the current analysis.

**The spatial variance would be important for trend slopes, for example if confidence intervals would be obtained from the trend. However, we agree, and have separated the discussion. Also, please see our short analytical argument that allows to derive a first-order estimate of the magnitude of the normalisation-induced biases. We hope the analytical argument/correction might be useful if observations up to the present would be compared for example with model simulations for the future.**

Page 4, lines 6-9: Over which time period where these alternative masks (2,3,4) defined? If not 1951-2010, you need to clarify that they may introduce the "regression to the mean" bias.

**The Köppen-Geiger classification is based on temperature and precipitation taken from the CRU TS 2.1 and the Global Precipitation Climatology Centre (GPCC), respectively, for the time period 1951-2000. Although this is not the full period, the period is (1) fairly long, and (2) two independent datasets are combined (temperature and precipitation), such that any potential "regression to the mean" effect should be negligible.**

Page 4, Line 9: What is the rationale behind including transitional regions when studying precipitation in dry regions?

**The rationale is simple: Our intention for this paper is to explore a range of different choices in order to test the sensitivity for different trend slopes and period increments of extreme precipitation – to this end, we believe that a combination of "arid" and "semi-arid" region can indeed provide additional insights.**

Page 4, lines 15/16: large parts of these 'subsidence regions' with no or little precipitation changes are located over the ocean. Water availability can clearly not be a limiting factor here, so this is unrelated to the discussion of different

definitions of 'dry'.

**We do not claim causality here – i.e. the statement does not imply that the reduced trend slopes in precipitation extremes in arid and semi-arid regions are due to water availability. This statement is just a short plausibility discussion of our results – given that the section is now entitled "Sensitivity of changes in precipitation totals and extremes to the definition of a dry region" we think this is appropriate.**

Page 4: Lines 17-21 give a hint of a balanced discussion, but unfortunately lead to a highly biased conclusion (lines 22-24), again appealing to what is supposedly 'commonly understood' and suggesting arid would be a conventional definition for dry.

**We have clarified and extended the conclusion: We report about the reduction in trend slopes, and indicate that there is a significant increase if the datasets are merged.**

**References**

Allan, R. P., Soden, B. J., John, V. O., Ingram, W. J., Good, P.: Current changes intropical precipitation. Environ. Res. Lett. 5, 025205, 2010.

Donat, M. G. , Alexander, L.V., Yang, H., Durre, I., Vose, R., Dunn, R., Willett, K., Aguilar, E., Brunet, M., Caesar, J., et al.: Updated analyses of temperature and precipitation extreme indices since the beginning of the twentieth century: the HadEX2 dataset, Journal of Geophysical Research: Atmospheres, 118, 2098—2118, 2013a.

Donat, M. G., Alexander, L. V., Yang, H., Durre, I., Vose, R., Caesar, J.: Global land-based datasets for monitoring climatic extremes, Bulletin of the American Meteo- rological Society, 94, 997-1006, 2013b.

Donat, M. G., Lowry, A. L., Alexander, L. V., O'Gorman, P. A., and Maher, N.: More extreme precipitation in the world's dry and wet regions, Nature Climate Change, 2016.

Greve, P., Orlowsky, B., Mueller, B., Sheffield, J., Reichstein, M., and Seneviratne, S. I.: Global assessment of trends in wetting and drying over land, Nature Geoscience, 7, 716—721, 2014.

Liu, C. and Allan, R. P.: Observed and simulated precipitation responses in wet and dry regions 18502100. Environ. Res. Lett. 8, 034002, 2013.

Sun, F., Roderick, M. L., Farquhar, G. D: Changes in the variability of global land precipitation. Geophys. Res. Lett.

39, L19402, 2012.

**Figure Caption** (complete caption as the online system seems to cut the cap- tion after the second sentence):

**Figure 1:** Extreme precipitation changes in arid and humid regions. Time se- ries of Rx1day (the annual-maximum daily precipitation) for dry/arid (a) and wet/humid (b) regions as identified by Greve et al., 2014. Area-weighted average time series are shown for HadEX2 and the ensemble mean and spread of CMIP5 simulations. Precipitation indices were first normalized by calculating annual values as a fraction of the 1951—2010 local mean before calculating the dry- and wet-region averages. Black lines, annual values from observations and ensemble mean; red lines, linear trend; blue dashed lines, 30-yr averages for 1951—1980 and 1981—2010; grey shading, ± one ensemble standard deviation. dRx1day indicates the difference between the averages during 1981—2010 and 1951—1980; slope is the linear trend Sen-slope estimate (unit, decade−1); and the p-value is the trend significance using a Mann—Kendall test. (c) The mask indicates the locations of the grid cells contributing to the average of the dry (red) and wet (blue) regions, and the number n of grid cells contributing to the area averages of dry and wet regions is given. Land grid cells that are less complete than 90

---

## Author Comment (AC5) · 21 Nov 2016

**Response to RC4:**

**S. Sippel et al., 2016**

Overall, I am pleased with the topic of the Sippel et al. paper, which is an evaluation and criticism of some of the methods used in the Donat et al. 2016 paper "More extreme precipitation in the world's dry and wet regions". This is the type of check-and-balance that keeps our science robust. Sippel et al. address two main criticisms of the Donat 2016 paper, (1) the introduction of a statistical bias when the rainfall data is normalised, and (2) the introduction of another statistical bias based on the regions that are selected as "dry" and "wet" regions. The overall flow and readability of the paper was dense, but not unfollowable. However, I understood the context of the paper, and the authors' intention, much better after I read the Donat et al. 2016 paper. The authors could use more precise wording to clarify that the methods used were done to recreate the results from Donat et al. 2016.

**We thank the reviewer for the positive evaluation of our manuscript. We agree that the original manuscript was partly very dense, and to address this issue we improve the embedding of Figures and Tables, provide better context, and add a second simple "illustrative example" to illustrate the "regression to the mean issue" a bit better. For more details, please see our reply to Reviewer #1, who raised a similar comment. We will also emphasize in a revised manuscript more precisely that the aim of our paper is to reanalyze the same dataset with a different methodology and choices for dryness definition in order to corroborate and test the sensitivity of the results.**

On the topic of the introduced bias from normalising the data; the process of normalising data is pretty common and ensures that areal averages are not dominated by very wet regions. However, this needs to be done with care. The authors unpack and clearly describe the statistical changes that are introduced from the normalisation process. I liked the illustrative example found on page three in lines six through 11 and the quantification of the bias (%) in appendices A and B provided good support for the argument. (Although it isn't clear why these are included as appendices and not tables in the paper). Furthermore, the authors do well to point out the changes that arise by using different reference periods to deconstruct the data (i.e. Figure 2). I note that it was not really clear from reading the Donat (2016) paper why they used the 1951—1980 period to normalise the data.

**Thanks again for the positive comments. We'll move the Appendices in the main manuscript and hope this will improve readability.**

**We agree with the reviewer that some normalization is often necessary to avoid that wet regions dominate spatial averages. Partly because this methodology is common (as pointed out by the Reviewer), please note that we have tried in addition to derive an analytical understanding / approximation of the biases. This will hopefully allow to estimate whether the systematic biases induced by reference period normalization in any particular case or study are worrisome or whether they are small and can be ignored. Please see the attached pdf-file, we intend to include this material in a revised manuscript as an Appendix or Supplement.**

I don't completely agree with the argument for selecting dry regions. The criteria and thresholds used to define a dry region are very subjective. As Sippel et al. point out, precipitation alone is not enough to determine if a region is wet or dry—e.g. at very high latitudes where even small amounts of rainfall can exceed the potential evapotranspiration. However, the criteria used are dependent on the question to be answered. If the question to be answered is, "How are global precipitation patterns changing?" then an analysis of precipitation alone would be sufficient. If you are trying to address, "Are wet/dry regions getting wetter/drier?" then the hydrology/aridity or climate classification of the region would need to be considered.

**We appreciate your comment: Please note that our intention was not to reject any particular dryness definition, but simply to explore the robustness of the results to this choice. However, we admit that this was not clear enough in the original manuscript, and we'll stress that both definitions can be appropriate depending on which question is being asked. Therefore, in a revised manuscript, we will refer to "regions with moderate extreme precipitation" (for Rx1d), "meteorologically dry regions" (with low precipitation totals), and to dry (arid) regions.**

The authors quantify the "regression to the mean" bias (as shown in appendices A and B) that arise by defining dry areas as the lowest 30%. The authors further demonstrate that by using the Köppen classification and the Greve (2014) definition that the large trends found by Donat et al. are dramatically minimised. I think this argument is a moot point because, as other reviewers have already pointed out, the HadEX2 dataset does not have data over the world's driest regions (e.g. the Sahara, Western Australia) or some of the wettest regions (e.g. the Amazon or the Maritime Continent region).

A global analysis or precipitation extremes or precipitation trends using HadEX2 data would deliver incomplete results.

**We agree that changes in precipitation characteristics as studied in our analyses are not representative or complete**

**given data-scarcity in many of the world's dry regions. However, we also believe that data-scarcity should not prevent scientific analyses being done with the data that is available at present. Therefore, we will emphasize this point clearly in the revised manuscript. In addition, please see our analyses in reply to Reviewer #2, where we have studied some of the characteristics of the data-scarce regions in more detail.**

Specific comments: 1. Page 4, line 12: mentions a two-sided trend test. Is this the same as the Mann–Kendall test used by Donat at al. and mentioned in the caption of figure 3? It is not really clear in the body of the text why or how this test was chosen.

**In the study of Donat et al. a one-sided Mann-Kendall trend test is used. Therefore, in all our figures and tables we report both one-sided (H0: No positive trend; value from the Donat et al. study are reproduced), and two-sided (H0: No trend) p-values. In a revised manuscript, we will phrase the text more in terms of the reduction of the trend slopes, rather than p-values only, because the latter can be misleading for relatively noisy time series (see e.g. Short comment by Donat et al. and our reply).**

2. Appendix A, Figure 0, caption: check the spelling of Köppen. This figure was hard to understand. After reading the caption a few times I understood that it is basically built as a table with the first (left) column being the PRCPTOT data and the second (right) column being the Rx1D data. It would be nice to have the rows/columns clearly labelled.

**Thanks for this point, we clearly see the need to improve the labels and caption and will do so.**

3. Figure 2: The caption mentions red lines. The lines look orange to me.

**Yes they are (erroneously), and they will be changed. Thanks for reading thoroughly.**

4. Figure 3: I found this figure very difficult to understand. There is a lot of information that is overlayed on other information. The grey text is too light against the white background.

**We will improve readability of Figure 3 by changing the colour and expanding the caption.**

5. Your methods for producing this graph (grey and black lines) are not clear. You mention the grey lines have been

corrected for "statistical artefacts"; I could not find this correction explained anywhere. Which artefacts have you corrected for? Is it the bias from the normalisation? Likewise, the process for producing the black line, or removing the incomplete data, is not explained.

**Yes, both the grey and black lines are produced by normalizing with the period means of the whole period, therefore avoiding the bias. Grey lines are based on the 90% completeness threshold in Donat et al., black lines are based on only 100% complete time series. We will improve this explanation in a revised manuscript.**

6. The label on the first row of graphs mentioned the Köppen–Geiger climate classification, but the caption references Köppen (1900). The Köppen–Geiger classifications were not published until Geiger (1954 and 1961). Kottek et al. 2006, which was mentioned in the text, is of the Köppen–Geiger classifications. Should the caption reference Kottek et al. 2006 rather than Köppen (1900)?

**Thanks for this hint! Yes, it should.**

7. Are graphs 3.e and 3.f from the Greve data, dry+transitional regions? It is not clear from the caption.

**Yes, they are. Thanks for reading thoroughly, we will add this to the caption.**

---

## Author Comment (AC6) · 21 Nov 2016

Jena, 21 November 2016

Dear Editors and Reviewers,

we thank four reviewers for their positive evaluation of our manuscript and their constructive comments. Further, we thank M. Donat for a partly positive, partly critical evaluation of our manuscript and constructive suggestions. We highly appreciate the scientific discussion that took place, which we believe is an important step to scrutinize and strengthen research findings. We respond to all comments in a point-by-point manner in the individual responses, and outline changes made to the manuscript. We

believe that the manuscript will improve significantly on the basis of the comments. Please note that we have attached an additional file (partly as a response to comments made by the Reviewers) that allows an analytical understanding and approximation of the normalisation-induced biases outlined in the manuscript.

Sincerely, Sebastian Sippel on behalf of all authors

Please also note the supplement to this comment:
http://www.hydrol-earth-syst-sci-discuss.net/hess-2016-452/hess-2016-452-AC6-supplement.pdf

[Figure]

**Supplement:**

**Appendix A: Analytical approximation of expectation value for the normalisation-induced bias**

Assumptions and Notation:

- Assume independent and identically distributed (i.e., stationary) variables $X_{t,i}$ with mean given by $\mathbf{E}(X) = \mu$ and variance $\mathbf{Var}(X) = \sigma^2$. Let the subscripts $t$ and $i$ denote time and grid cell index, respectively.

- Denote $t$ as an arbitrary time step in the 'out-of-base' (independent) period, and $t_{ref}$ as an arbitrary time step inside the reference period. Let $n_{ref}$ denote the length of the reference period.

- Denote $\Delta_{bias} = \mathbf{E}(\frac{X_{t,i}}{\hat{\mu}_{\text{ref,i}}}) - 1$ as the change induced by normalisation by the mean of an independent reference period (i.e., 'normalisation bias').

Our objective is to find an analytical approximation of the expectation value for the artificially induced relative change ('bias') by dividing a random variable $X_{t,i}$ as defined above by its sample mean ($\hat{\mu}_{\text{ref,i}} = \frac{1}{n}\sum_{t_{ref}=1}^{n_{ref}} X_{t_{ref},i}$, where $\mathbf{E}(\hat{\mu}_{\text{ref,i}}) = \mu$) that has been estimated in an independent ('reference') period, i.e.

$$\Delta_{bias} = \mathbf{E}(\frac{X_{t,i}}{\hat{\mu}_{ref,i}}) - 1 \approx f(\mu, \sigma, n_{\text{ref}}). \tag{A1}$$

Clearly, an unbiased estimate of this quantity in stationary time series should yield $\Delta_{bias} = 0$. Because $X_{t,i}$ and $\hat{\mu}_{
[revised manuscript text omitted]

Similarly to above, we find that the ratio of location to scale parameter ($\frac{\mu'}{\sigma'}$), for any fixed reference period length ($n_{\text{ref}}$), determines the magnitude of the bias. The analytical approximation can be verified by numerical simulation using GEV-distributed random variables, and is found to fit the data very well (Fig. A-2a). Furthermore, an estimator of the expectation value of the biases by only estimating the mean and standard deviation of empirical time series (i.e., using the first term in the Taylor approximation) can be derived easily and is found to work reliable also for a small number of independent grid cells (Fig. A-2c).

**ii. GEV distribution with $\xi \neq 0$**

Here, we test whether the analytical argument from above can be extended to Generalized Extreme Value distributions with $\xi \neq 0$. In practical applications of the GEV to observed maximum precipitation, a shape parameter of $\xi \approx 0.2$ is often found

(Van den Brink and Können, 2011), therefore we test here for $X_{t,i} \sim \text{GEV}(\mu', \sigma', \xi = 0.2)$. The expectations for mean ($\mu$) and variance ($\sigma^2$) of a GEV, when $0 > \epsilon < 1$, are given by $\mu = \mu' + \sigma' \frac{\Gamma(1-\xi)-1}{\xi}$ and $\sigma^2 = (\sigma')^2 \frac{(g_2 - g_1^2)}{\xi}$, where $g_k = \Gamma(1 - k\xi)$, $k = 1, 2$, and $\Gamma(t)$ is the gamma function (Johnson et al., 1995).

Hence, the (dominant) quadratic term in the Taylor approximation in Eq. A7 reads,

$$\Delta_{bias} \approx \frac{(g_2 - g_1^2)}{n_{\text{ref}} \xi [\frac{\mu'}{\sigma'} + \frac{\Gamma(1-\xi)-1}{\xi}]^2}. \tag{A15}$$

The approximation works again very well in numerical simulations (Fig. A-2b), and shows that if $\xi \neq 0$, there is a dependency on $\xi$, $n_{\text{ref}}$, and again the ratio of $\frac{\mu'}{\sigma'}$ (rather than either $\mu'$ or $\sigma'$ individually), which determine the magnitude of the normalisation-induced bias. Please note that the approximation works similarly well for random variables drawn from a GEV-distribution with negative shape parameter ($\xi = -0.2$, not shown).

**Short Remark on in-stationarities in the out-of-base period**

Many real-world precipitation time series show in-stationarities due to climatic variations (O'Gorman, 2015) that are typically disgnosed as relative changes in the precipitation amount. Hence, the question whether and how any 'real change in the expectation value' outside the reference period can be disentangled from normalisation-induced biases becomes topical. Given the analytical approximation above, it becomes obvious that the highlighted normalisation-induced bias scales in-stationarities in the out-of-base period in a multiplicative way:

Let $c$ denote any change between the reference period expectation and some future period (e.g. assume one is interested in global or latitudinal changes in a past and future climatic period), i.e. such that $\mathbf{E}(X_{t_{\text{ref}},i}) = c\mathbf{E}(X_{t,i})$, then the bias ($\Delta_{\text{bias}}$, after accounting for the 'real change') would simply scale with the relative change ($\Delta$ denotes the total apparent change):

$$\Delta = c\mathbf{E}(\frac{X_{t,i}}{\hat{\mu}_{ref,i}}) - 1 \tag{A16}$$

$$= c\mathbf{E}(\frac{1}{1 + \epsilon_i}) - 1 \tag{A17}$$

$$= \underbrace{c - 1}_{\text{true change}} + c[\underbrace{\frac{\sigma^2}{\mu^2 n_{\text{ref}}} - \mathbf{E}(\epsilon_i^3) + \mathbf{E}(\epsilon_i^4) - \mathbf{E}(\epsilon_i^5) + ...]}_{\Delta_{\text{bias}}} \tag{A18}$$

From Eq. A18, it is straightforward to see that for any multiplicative changes in the expectation of the out-of-base variables, the normalisation-inudced bias scales with the change. Hence, this expression implies that to detect the 'true change $c$' between two periods, the normalisation-induced bias has to be accounted for, i.e.

$$c = \frac{\Delta + 1}{1 + \Delta_{\text{bias}}}. \tag{A19}$$

Numerical simulations can be easily conducted similar to Subsection 1.1 and 1.2 to verify that this scaling holds (not shown).

[Figure]

**Figure A-2.** a) Ratio of location to scale parameter vs. normalisation-induced bias in a Generalized extreme value distribution for numerical simulations with various location parameter values (dots) and a) zero shape parameter, and b) with $\xi = 0.2$. Reference period length is taken as $n_{\text{ref}} = 30$, and numerical simulations are conducted with $n = 10^5$ grid cells with each 60 time steps. c) Variation in the empirical estimates of the biases (darkblue) for a given number of independent grid cells ($\frac{\mu'}{\sigma'} = 1$, $\xi = 0$, $n_{\text{ref}} = 30$). The magnitude of random changes in stationary time series with $n_{\text{ref}} = 30$ and $n_{\text{obase}} = 30$ is shown for comparison in black. Error bars indicate 5th and 95th percentile in repeated numerical simulations.